# Dealing with mixed and non-normative traffic. An agent-based simulation with the GAMA platform

Arnaud Saval[1], Duc Pham Minh[2,3], Kevin Chapuis[6], Pierrick Tranouez[1] *, Clément Caron[4], Éric Daudé[4], Patrick Taillandier[2,3,5]

**1** UR LITIS, University of Rouen Normandy, Rouen, France, **2** UMI UMMISCO, IRD, Sorbonne University, Bondy, France, **3** LMI ACROSS, Thuyloi University, Hanoi, Vietnam, **4** UMR IDEES, CNRS, Normandy University, Rouen, France, **5** UR MIAT, INRAE, University of Toulouse, Castanet-Tolosan, France, **6** UMR 228 ESPACE-DEV, IRD, University of Montpellier, Montpellier, France

◔ These authors contributed equally to this work.
* Pierrick.Tranouez@univ-rouen.fr

**Data Availability Statement:** the code of the model and of the data are available here: https://doi.org/10.6084/m9.figshare.21369090.v1.

## Abstract

Continuous improvement in computing power allowed for an increase of the scales micro-traffic models can be used at. Among them, agent-based frameworks are now appropriate for studying ordinary traffic conditions at city-scale, but remain difficult to adapt, especially for non-computer scientists, to more specific application contexts (e.g., car accidents, evacuation following a natural disaster), that require integrating particular behaviors for the agents. In this paper, we present a built-in model integrated into the GAMA open-source modeling and simulation platform, allowing the modeler to easily define traffic simulations with a detailed representation of the driver's operational behaviors. In particular, it allows modelling road infrastructures and traffic signals, change of lanes by driver agents and less normative traffic mixing car and motorbike as in some South East Asian countries. More-over, the model allows to carry out city-level simulations with tens of thousands of driver agents. An experiment carried out shows that the model can accurately reproduce the traffic in Hanoi, Vietnam.

## 1 Introduction

Modeling traffic is about as old as car traffic itself [1]: the first ideas date back to the 1920s (like Frank Knight traffic equilibrium) and the first models to the 1950s [2]. Based on the best methodology available at the time, they mirrored the practice of physics and in particular, fluid mechanics. The reasoning was the same: although it was known that traffic was composed of cars, like gases were composed of molecules, with their own set of defining characteristics, it was more productive to consider the resulting aggregations like mean speed, flow or density. Principles of conservation of mass and energy could then lead to the proof of algebraic and differential equations describing the links between these variables. These models would later be known as *macroscopic*.

**Funding:** This work is funded by the ANR ESCAPE project, grant ANR-16-CE39-0011-01 of the French National Research Agency (https://anr.fr/) The funders had no role in study design, data collection and analysis, decision to publish, or preparation of the manuscript.

**Competing interests:** The authors have declared that no competing interests exist.

These analytical models focused on the study of traffic phases (in terms of thermodynamics): at first two (free flow and congested), then three phases (free flow, wide jam and synchronized) [3]. Keeping these equations solvable or step-based and approximable requires the features modeled to be kept sparse: the geometry of the road is generally ignored, it is often a single road composed of a single lane, with equations that model the collective evolution of vehicle described by average speed, average behavior, average characteristics. These macroscopic models reproduce the shock waves corresponding to vehicle braking and acceleration with the decrease in the density of vehicles usually observed in the traffic. These models are also able to reproduce the phenomenon of congestion. However, by their nature, they do not allow to study the individual behavior of vehicles and neglect the diversity of drivers in the flow. These models therefore do not allow the study of individual or collective road traffic practices that do not follow either international conventions (never drive against the traffic flow) or national regulations (only one vehicle per lane).

In the 1970s, some diversity was provided to the vehicles/particles in models often inspired by gas kinetics [4]. These models were often stochastic: a velocity statistical distribution is generally computed, but the individuality of each vehicle is not really modeled. These *mesoscopic* models are not individual-based, as set out by [5] (e.g. [6]).

To make the drivers and vehicles more individualized, *microscopic* models were developed. These models process the diversity and heterogeneity of the vehicles and their environment in real terms. Although some are equational [7, 8], the development of computers and computer sciences led to more complex computer models that used cellular automata at first and then, agent-based systems to treat drivers and vehicles as individual units instead of a continuous flow or particles.

Agent-based models (ABM) enable to describe each driver as an autonomous agent, making decisions based on its own aptitudes and local environment. It's then possible to model how each vehicle moves according to surrounding infrastructure and other vehicles, accelerates and slows down according to its environment, while maintaining a safe distance with other cars. Thus, the simulated traffic results from the following of rules and regulations by each driver, the organisation of road infrastructures and interaction of each agent with the other road users. These capabilities make this modeling approach particularly appropriate for studying, at a fine scale (spatial and temporal), the impacts of uncommon events (road closure, major accident) on the traffic. Indeed, in such a context, being able to simulate the traffic in a realistic way while taking into account the road infrastructure (crossings, traffic signals, etc.), the properties of the cars (length, maximum speed, etc.) and the personality of the drivers (tendency to respect the norms), is mandatory. Equational models inputs can be used for in-lane car-following model, as for example the Intelligent Driver Model [9] transformed to be used in an agent-based model traffic simulation [10].

Even if there are many frameworks nowadays dedicated to the development of agent-based traffic models, several models are still developed from scratch or by using a generic platform (e.g. [10–12]). Indeed, if the existing frameworks allow to study many aspects of traffic (vehicle flow at city-scale, the movement of vehicles on a roundabout. . .) in "normal" conditions, they cannot be easily adapted by domain experts who are not necessarily computer experts to study specific problems, for example, integrating an emotional dimension in the behavior of agents in a crisis situation.

In this paper, we present a new built-in model integrated into the open-source GAMA modeling and simulation platform [13, 14] dedicated to the development of fine-scale traffic simulations. GAMA provides modelers who may or may not have computer-programming skills with tools to develop complex models. In particular, it offers a complete modeling language (GAML: GAma Modeling Language) and an integrated development environment that

allows modelers to build models quickly and easily. The model developed allows GAMA users to easily define traffic simulation at a fine scale, with a detailed representation of the driver operational behaviors. We describe in this paper a recent addition to GAMA's traffic module able to take into account mixed modes of non-normative traffic. Indeed, the traffic rules, or rather the way the drivers appropriate them, can be very different from one country to another: people do not drive the same way in France as in Vietnam where motorcycles are in the majority and the driving rules are much less followed. Our integrated model takes the form of a plugin delivered with the GAMA platform (version 1.8.2). This plugin allows modelers to attach to their agents new attributes and actions related to traffic simulation (concept of *skill* in GAMA). It also provides a set of simple models to illustrate the use of these new features. The biggest advantage of integrating this traffic model into GAMA rather than into a dedicated traffic framework such as SUMO is that it facilitates the enrichment and coupling of this model. Indeed, unlike other traffic simulation systems that require separate projects or external tools, this extension for GAMA allows advanced driving and agent behaviors to be defined with the same language. So, for example, while adding new elements such as vehicle-pedestrian interaction or emotional dimensions to agents is complicated in frameworks like SUMO, it can be done very simply with GAMA (which has already integrated many models). For the same reasons, the integration in GAMA allows to easily couple this traffic model to other models (urban dynamics, epidemiology, disaster management).

This paper is organized as follows: Section 1 presents the related works, in particular, the existing agent-based traffic simulators and frameworks. Section 2 is dedicated to the presentation of the integrated model developed. Section 3 present elements of the model's validation through the application for the simulation of the traffic in Hanoi, Vietnam. Finally, Section 4 concludes and presents some perspectives.

In order to follow the open-science movement and make our results as replicable as possible, the code of the integrated model is provided with GAMA (version 1.8.2): it is thus open-source and easily downloadable from the GAMA Website. In addition, the GAML models used for the experiment are published on FigShare with their geographic data and we present all the necessary parameter values in this paper [15].

## 2 Related works

As stated before, we put emphasis on freely re-usable open-source framework to build agent based traffic model. Hence, we excluded from this brief state of the art, most of the well known commercial frameworks that populate the research landscape, like VISSIM, AnyLogic, Paramics or Aimsun to only mention a few. In this section, we rather focus on open source traffic simulation frameworks that have been developed and released in the last two decades [16]. We briefly review the most influential ones and urge readers to look at dedicated review papers to find out more comprehensive information, e.g [17]. We split the review of tools into two categories, agent-based framework purposefully dedicated to traffic modeling (Page 3) and generic agent-based modeling framework that are often used to tackle traffic modeling related issues (Page 5).

### 2.1 Agent-based framework dedicated to traffic modeling

MATSim [18] or Multi Agent Transport Simulation Toolkit is an open-source (GPL) Java application consisting of several modules that can be combined. MATSim follows an activity-based paradigm and proposes many advanced features dedicated to traffic simulations that can be enriched by users through the definition of new modules in JAVA. Among these features, those related to daily mobility and multimodality (automobiles, public transport, walking,

cycling) enable the simulation of millions of synthetic agents. The allocation of new attributes (e.g. having a bicycle or not, preferences for and availability of public transport) required for agents to use additional features implies at least to modify the synthetic population generation process. While the platform has been used to model traffic under crisis context (e.g. [19]), it remains a mesoscopic simulation framework, and as such no extensions has been brought to systematically deal with non normative driver behavior and mixed interacting modes.

TRANSIMS [20] is also an activity-based travel model where agents make travel decisions according to an activity schedule. Four distinctive modules exist in TRANSIMS: Population Synthesizer, Activity Generator, Route Planner and Microsimulator. Activity generator use household survey data to work out scheduled dimensions of activity patterns of each individuals generated by population synthesizer. During simulation, a feedback mechanism between Router and Microsimulator brings the system into equilibrium, essentially by modifying the scheduling inside initial plans. While the framework has attracted a lot of interest in the early 2000s, it has now been dropped off and no new addition has been made since, with no feature included to model non normative traffic behavior and multi-modality.

SUMO [21], another popular open-source framework is a suite of applications to help modelers prepare and perform traffic simulations. Like MATSim, it proposes many advanced features dedicated to traffic simulations that can be enriched using C++ language. SUMO is used in particular to optimize the positioning of counting loops on the road network, or for the improvement of traffic light phasing in order to minimize traffic congestion. It provides tools to generate vehicles schedules and it must be integrated with third party application to support complex pedestrian behaviors.

SUMO was designed to simulate traditional European and American traffic. Nevertheless, [22] have proposed to adapt it to Chinese traffic: in order to take into account motorcycles, [22] have developed an extension allowing to divide the road lanes into sub-lanes—each vehicle occupying one or more sub-lanes. Yet, except for overtaking behavior, this extension does very little to take into account the non-normative driving behavior: for instance, vehicles cannot use wrong way roads or disrespect red traffic lights. Concerning this point, a dedicated extension was recently proposed to monitor and control non-normative traffic events in the context of traffic with autonomous cars [23]. However, this extension only concerns the computation of vehicle speed (and acceleration, braking) and not their behavior with respect to traffic rules in general (traffic lights, right/left priorities, etc.). Moreover, SUMO provides only limited interactions between agents in traffic situations (e.g., a pedestrian can only cross the street at a red light), which makes it very difficult to represent crisis and non-normative traffic contexts.

SimMobility [24] is a recent open-source framework written in C++ specifically targeting simulation of prospective mobility patterns in urban areas. It has been built upon 3 sub-models, ranging from short (lower than a second), middle (minutes to hours) to long-term (from days to years) dynamics, hence it is able to represent a variety of mobility choices, including micro (e.g. speed regulation, lane change) and macro-level traffic decisions (e.g. path finding, traffic jam avoidance) and higher order agent decisions on home place and workplace locations, choices of mobility modes and more. As it is the case for the previous platforms, SimMobility does not incorporate natively non normative driving behavior patterns (i.e. the micro-driving decision is based on the MITSIM engine [25]), whereas mixed traffic modeling remain extremely limited, with each mode mobility being modeled separately.

Other dedicated open-source frameworks have been proposed (e.g. AgentPolis—[26], Arena—[27], Polaris—[28]), but to the best of our knowledge, none of them offers a native integration of non normative mixed traffic. One of the main criticism comes from the use of such platform whose first intent is to model regular urban traffic context. By design, they

might not be suitable studying extra-ordinary or emergent traffic situation [29]. In this regard, generic purpose agent-based modeling platform can easily bring such feature on top of regular traffic models. In the next sub-section we briefly outline strengths and weaknesses of generic purpose agent based modeling tools to represent non normative mixed traffic.

## Generic agent-based framework used in traffic modeling

Although the traffic oriented frameworks are powerful and propose many advanced features (see Table 1 of [30] for a structured comparison of traffic simulators), they hardly deviate from their original purpose of regular western traffic modeling. Another main issue is the accessibility of such specialized tools: for modelers without high-level programming skills, adapting these platforms to specific application contexts is not possible, as they are required to write code in JAVA or C++. All those platforms put emphasis on particular aspects of transport system (e.g. traffic optimization, future mobilities) and rely on plugins and additions to their core model to explore related issues. In this regard, general purposes agent based modeling frameworks grant easy access to model building and make it possible to quickly adapt transportation model to related issue of concern, like mass evacuation [31] or pollution emission [32].

However, in the case of generic modeling and simulation platforms, only a few propose tools that can be used to develop traffic simulations. One of these, Repast Symphony [33] proposes interesting features with respect to GIS loading and graphs. Using this platform to develop complex models requires writing code in JAVA. For prototyping models, which do not need to integrate several layers of heterogeneous geographic information (networks, polygons, raster) and which can be simulated with a few thousand agents, the NetLogo software is a good compromise due to the wide choice of available libraries and the ease of the programming language [34].

The GAMA platform [14] also provides various features that can be used by modelers to develop traffic models. In particular, GAMA features GIS data loading (shape files, OSM data...), graphs defined from polyline geometries, shortest paths computation and agents movements on polyline networks. If these features are appropriate for the development of traffic simulations at a large time-scale (see for example the MIRO project [12] with a time scale of 10 minutes per step), they do not allow one to simulate driver behaviors at a fine scale: change of lanes, effect of traffic signals, etc.

To overcome these limitations, we developed an integrated traffic model in Java directly operational in a model developed in GAML. This model provides a set of classical concepts (road, intersection, driving agent) and actions (driving...), with an ability to model mixed traffic (car, motorcycle, bicycle...) and non-normative behaviors (in which traffic rules can be ignored). Another specificity is the possibility to implement feedback processes, such as activity rescheduling, due for example to a major disruption scenario. In this case, many agents will need to cancel their upcoming activities (target) and adopt a new agenda with partial awareness of the environmental context and of other agents decisions. We base this model on two classical sub-models: *IDM* (Intelligent Driver Model) for the computation of the vehicle speed [9] and *MOBIL* (Minimizing Overall Braking Induced by Lane change) for defining when changing lane [35].

## 3 The GAMA integrated traffic model

### Structure of the road network

One important aim of the integrated model is to allow simulation at the scale of a city using ubiquitous data, i.e. available for most cities in the world, such as OpenStreetMap (OSM) data

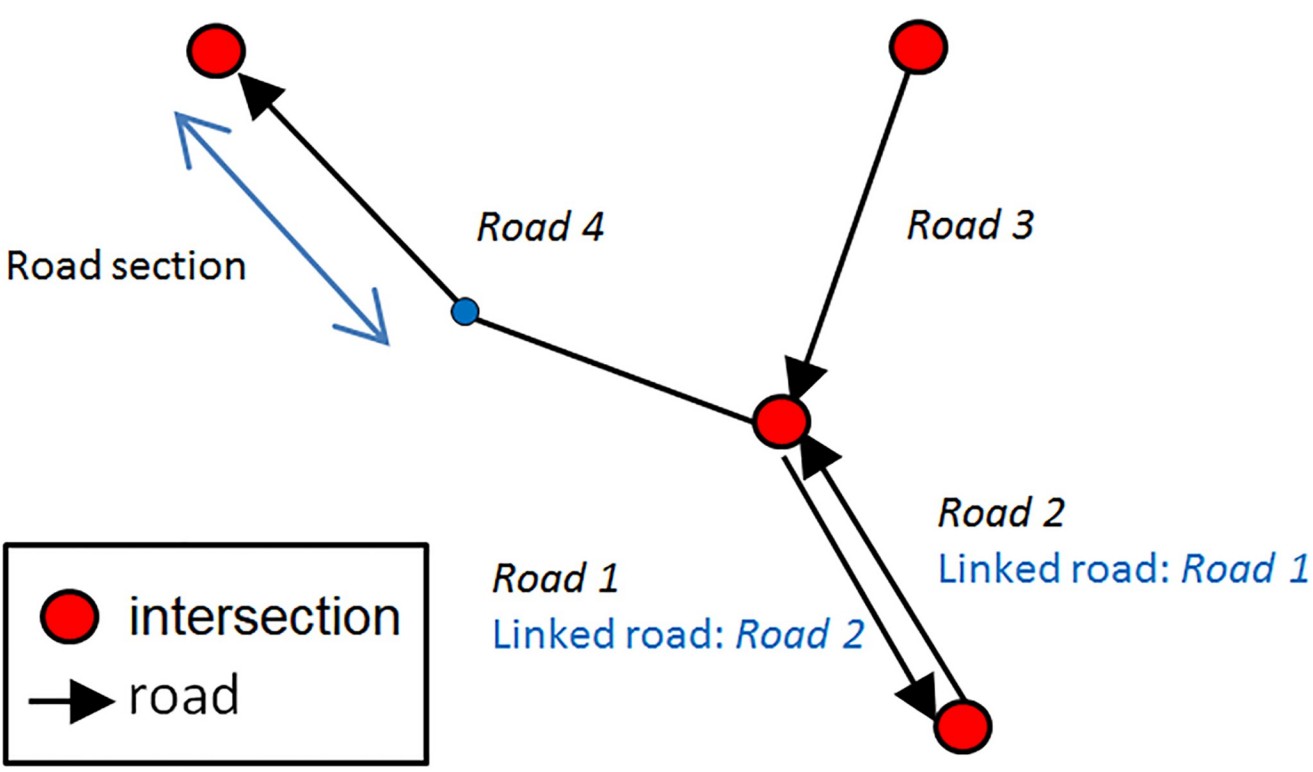

**Fig 1. A road network composing of roads and intersections.**

for example. Therefore we choose to use a classic format for the roads and intersections, as illustrated in Fig 1.

Each road is a unidirectional edge represented as a polyline composed of segments. Each road is linked at each of its two ends to an intersection. Two-way roads will be represented by two roads. In order to allow a vehicle on a bi-directional road to use the road in the opposite direction to overtake another vehicle, we integrate the notion of linked road. A linked road in relation to a road is a road in the opposite direction that a driver can use to overtake another vehicle.

We also integrate the notion of lanes: a vehicle will be able to change lanes and even use the lanes of the linked road (see Page 6). In traffic law, a lane is generally a sub-division of a road which can be used by only one vehicle in its width [22]. Vehicles can swith lanes depending on the road markings and the willingness of the agents to respect these. Our model can reproduce that if that is appropriate for the modeled city. There are other cities, like for example Vietnamese cities, where the width of the whole road is used by as many vehicles as can fit, from bicycle to trucks. Rather than going down a too-detailed geometric modelling, we model this kind of traffic by introducing the concept of sub-lanes. A sub-lane is a division of a road where can fit only one of the smallest vehicle modelled, bicycles/light motorbikes in our Hanoi example. A bigger vehicle uses up several sub-lanes at once, up to the total number allowable on the considered road. Vehicles can switch sub-lanes as space allows.

Another important property is the road speed limit, which can also be provided by OSM data. This variable is a key element for the definition of vehicle speeds. We choose to allow agents to modify this value during the simulation, for example to take into account the fact

**Table 1. Attributes of the road entities.**

| Name | Type | Description |
|---|---|---|
| lane number ($l_{number}$) | integer | Number of lanes of the road |
| speed limit ($v_{roadLimit}$) | float | Speed limit of the road |
| linked road ($r'$) | road | Linked road if this road is a bidirectional road |
| source intersection ($s_i$) | intersection | Intersection which the road leads out from |
| target intersection ($t_i$) | intersection | Intersection which the road leads to |

that, after an accident, a lane of a road is closed or that the speed limit of a road is decreased by the authorities.

Table 1 presents the attributes of the road entities.

Intersections will be characterized by their incoming and outgoing roads. They can also be used to simulate stop signs or traffic lights and to take into account the fact that some drivers can block an intersection.

Table 2 presents the attributes of the intersection entities.

## Modeling vehicles

A vehicle is first characterized by its location *loc*, a 3D-point (coordinate) that represents the centroid of the vehicle. The actual geometry of the vehicle is not taken into account. However, the size of a vehicle is determined by its length and the number of lanes it occupies. In our Vietnamese example where the lanes are defined according to the size of the motorcycles, we can consider that a motorcycle will occupy one lane, but that a car will occupy two.

The lanes which are occupied by a vehicle are defined by its lowest lane index in conjunction with the number of lanes it spans, as illustrated in Fig 2. The list of attributes related to the location of the vehicle in space are shown in Table 3.

Before being able to move, the vehicle must first invoke the **compute_path** method to initialize some internal variables and in particular its path to follow. This method takes a road graph and a target intersection as inputs, and then finds the shortest path to the target. GAMA provides several algorithms to carry out this computation such as Dijkstra, Bellman Ford, Floyd Warshall, A*, NBA* [36]. The driving variables initialized by the `compute_path` function are presented in Table 4.

The attribute *undirected* can be toggled to make the graph undirected during the shortest path computation, allowing drivers to move in the wrong direction on certain roads. This can be used for simulating pedestrians mobility (who are allowed to walk in both directions on the same road), or reckless drivers.

**Table 2. Attributes of the intersection entities.**

| Name | Type | Description |
|---|---|---|
| roads in (*in*) | list of roads | List of roads that lead to this intersection |
| roads out (*out*) | list of roads | List of roads that lead out of this intersection |
| regulators (*stop*) | dictionary | Dictionary that maps priority regulators (stop signs, traffic signals etc.) to the list of roads it regulates (more details about this attribute is be given Page 6) |
| blocked drivers (*bd*) | dictionary | Dictionary that maps a driver to a list of roads, indicating which roads are currently blocked by that driver (the purpose is covered Page 6) |

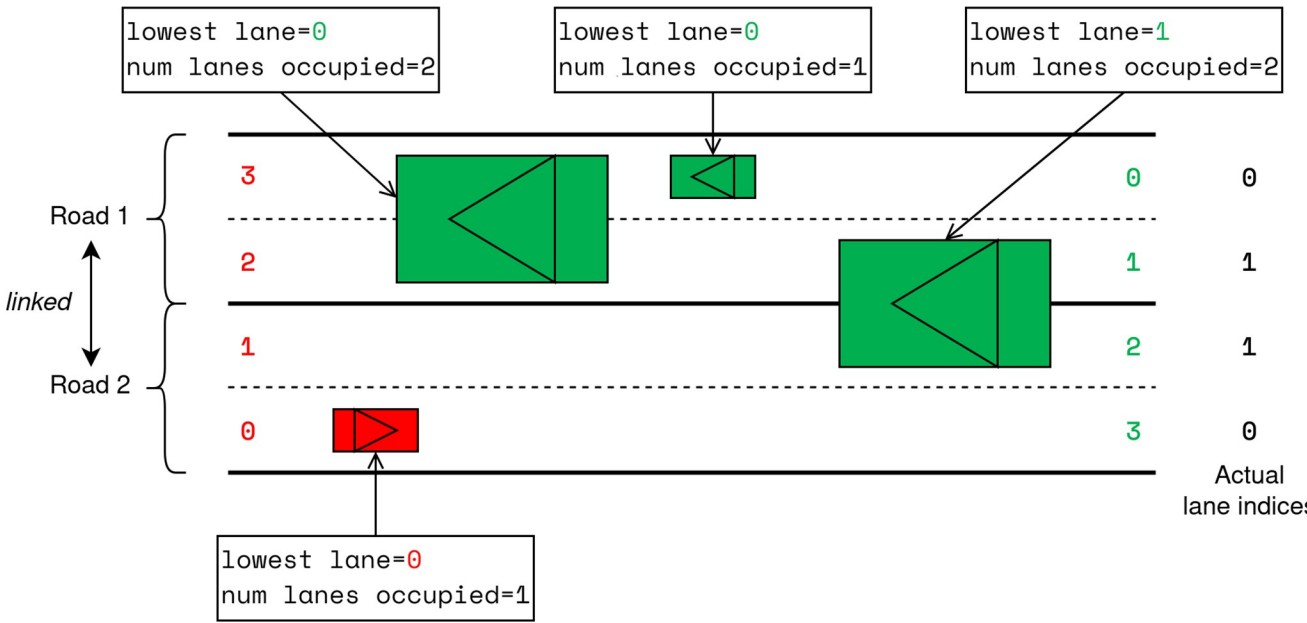

**Fig 2. The "lane states" of several vehicles driving on two linked roads.** Note that the right-most vehicle is occupying lanes with indices 1 and 2 from its point of view, but the latter is in fact lane 1 on the linked road.

**Table 3. Attributes of the vehicle entities related to spatial location.**

| Name | Type | Description |
|---|---|---|
| Location (*loc*) | 3D coordinate | Coordinate of the centroid of the vehicle |
| Vehicle length (*len*) | Float | Length of the vehicle (in meters) |
| Lowest lane index ($l_{lowest}$) | Integer | Index of the lowest lane occupied by the vehicle |
| Number of lanes occupied ($l_{occupied}$) | Integer | Intermediate intersection in the path which the driver is heading towards |

**Table 4. Attributes of the vehicle entities related to path initialization.**

| Name | Type | Description |
|---|---|---|
| Current path (*path*) | List of roads | List of roads that lead to the final destination |
| Current index (*index*) | Integer | Index of the road that the driver is currently on |
| Current target (*target*) | Intersection | Intermediate intersection in the path which the driver is heading towards |
| Final target (*f_target*) | Intersection | Final destination in the path |
| Ignore road direction (*undirected*) | Boolean | Can the vehicle take roads in the wrong direction to reach its destination? |

After variables initialisation, the vehicle can use the `drive` method (Algorithm 1), which takes no argument, in every simulation step to move across the road network and towards the target intersection.

**Algorithm 1** `drive` function

```
1: function DRIVE
2:     t ← simulation step duration
```

```
 3:   while t > 0 do
 4:     if driver at final intersection in path then
 5:       t ← 0          ▷Arrived at destination
 6:     else if driver at intermediate intersection in path then
 7:       n ← intermediate intersection
 8:       r ← next road in path
 9:       if readyToCross (n, r) then
10:         (l, a_c) ← chooseLane(r)
11:         v ← computeSpeed(a_c)
12:         if v = 0 & random(0.0, 1.0) ≤ p_block_node then
13:           t ← 0          ▷Blocked in the intersection
14:         else
15:           t ← updateDriverPosition(r, v, l, t)
16:         end if
17:       else
18:         t ← 0          ▷Blocked at the intersection
19:       end if
20:     else
21:       r ← current road in path
22:       (l, a_c) ← chooseLane(r)
23:       t ← updateDriverPosition(r, v, l, t)
24:     end if
25:   end while
26: end function
```

The overall driving process is summarized using pseudo-code in Algorithm 1. The provided algorithm features several functions: **readyToCross**, **chooseLane**, **computeSpeed** and **updateDriverPosition**.

the **readyToCross**($n$, $r$) function that checks several conditions to see if it is possible for the driver to cross the intersection $n$ before entering a new road $r$:

- There is no vehicle blocking the intersection;

- There is no red traffic light nor stop sign;

- There is no vehicle coming from the right (or left if the agent drives on the left side) at a distance lower than its safety distance.

For the second and third conditions, the driver can violate the rules and ignore them, as modelled by probabilities $p_{respect\_stops}$ and $p_{respect\_priorities}$. However, even if these conditions are fulfilled and the vehicle is able to cross the intersection $n$, it is possible that the next road $r$ is highly congested and there is no space left to enter the road (line 12 in Algorithm 1). In this case, the driver can choose not to cross the intersection just yet to avoid blocking vehicles coming from the sides, or just block the intersection with probability $p_{block\_node}$. Indeed, if a vehicle arrives at an intersection and wants to continue on the next road, on which one or several vehicles block the way (i.e. there is no room for new vehicles to enter this road), it may choose to stick the vehicle in front of it even if it blocks the intersection for drivers coming from perpendicular roads. In this case, the vehicle will update the *stop* attribute of the intersection: the key of the map will be the vehicle, and the associated values, the perpendicular roads.

The **chooseLane**($r$) function allows the vehicle on the road $r$ to determine both the best lane ($l$) and its acceleration ($a_c$) on this lane. The algorithm 2 describes this function which is based on the models *IDM* and *MOBIL* (see more details after).

The **computeSpeed**($a_c$) function computes the new speed $v$ of the vehicle according to its current speed, its acceleration $a_c$ and the maximal speed of the vehicle $v_{max}$ (see Eq 1).

Finally, the function *updateDriverPosition(r, v, l, t)* updates the vehicle's position on the road *r* based on its new lane *l*, speed *v*, and the time remaining for the vehicle to move during the current simulation step *t* and returns the time remaining after this move. Indeed, if the vehicle reaches the end of the road before the time has elapsed, it can continue after its movement on another road in the same simulation step.

To simulate how drivers accelerate and decelerate, we implemented the Intelligent Driver Model [9], which is a widely used car-following model. In this model frame, the acceleration of a vehicle is determined by its leading vehicle, i.e. the closest vehicle ahead which shares at least one lane with it. Eq 1 describes how the model computes the acceleration of a vehicle ($dv/dt$—corresponding to $a_c$ in Algorithm 1) from four input variables: the speed ($v$) of the vehicle, the speed difference between itself and the leading vehicle ($\Delta v = v - v_{\text{lead}}$), the bumper-to-bumper distance to the leading vehicle ($s$), and the speed the vehicle would drive at in free traffic ($v_0$).

$$\frac{dv}{dt} = a \left[ 1 - \left( \frac{v}{v_0} \right)^{\delta} - \left( \frac{s^*(v, \Delta v)}{s} \right)^2 \right] \qquad (1)$$

where $s^*(v, \Delta v)$ is the "desired dynamical distance", which is computed as follows:

$$s^*(v, \Delta v) = s_0 + \max \left[ 0, \left( vT + \frac{v\Delta v}{2\sqrt{ab}} \right) \right] \qquad (2)$$

The computation of $v_0$ on a road $r$ depends on the road speed limitation ($v_{roadLimit}(r)$) and on the tendency of the driver to drive under or below the road speed limitation ($coeff_v$).

$$v_0(r) = coef_v \times v_{roadLimit}(r) \qquad (3)$$

All the attributes that are related to the speed computation are described in Table 5.

If no leading vehicle is found on the next road segment, we will conclude that there is no leading vehicle at all, and *s* is set to $\infty$, making the third term in Eq 1 vanishes and hence allowing the vehicle to reach its maximum acceleration. To simulate red traffic lights, if the driver chooses to respect it (with the probability $p_{respect\_stops}$) then the road node containing the traffic lights is treated as a vehicle of length zero. When the lights turn green (or if the vehicle chooses to not respect it), the intersection will be considered as an intersection again.

Once the acceleration has been found by *IDM*, the vehicle's speed is updated by computing:

$$v_{\text{new}} = \min(v_{\text{old}} + \frac{dv}{dt}\Delta t, v_{max}), \qquad (4)$$

**Table 5. Attributes related to speed computation.**

| Name | Type | Description |
|---|---|---|
| Speed coefficient ($coef_v$) | float | tendency to drive under or below the road speed limitation. A value of 1.0 means that the driver will always try to drive at the road speed limitation. |
| Maximum speed ($v_{max}$) | float | Maximum speed that the vehicle can achieve |
| Time headway ($T$) | Float | Minimum time difference between the current vehicle and its leading vehicle when they pass a given point |
| Minimum gap ($s_0$) | Float | Minimum bumper-to-bumper distance to the leading vehicle that the current vehicle must maintain |
| Acceleration ($a$) | Float | Maximum acceleration of the vehicle |
| Deceleration ($b$) | Float | Comfortable braking deceleration of the vehicle |

in which $v_{\text{new}}$ and $v_{\text{old}}$ are the updated and old speed respectively, $\Delta t$ is the simulation step duration, and $v_{max}$ the maximal speed that a vehicle can achieved.

The vehicle location *loc* is updated using the formulae:

$$loc_{\text{new}} = loc_{\text{old}} + loc_{\text{old}}\Delta t + \frac{1}{2}\frac{dv}{dt}(\Delta t)^2. \tag{5}$$

Regarding how drivers move between lanes, we use the MOBIL lane-changing model [37] based on the Intelligent Driver Model. The model provides *two* criteria governing when a driver should change a lane during driving: the safety criterion (is it safe for the vehicle behind?) and the incentive criterion (is the new lane more attractive?).

The safety criterion is stated as:

$$\tilde{a}_{\text{n}} > -b_{\text{save}}, \tag{6}$$

in which $\tilde{a}_{\text{n}}$ is the acceleration of the new follower if the driver does change to the corresponding new lane. This criterion ensures the lane change does not cause the new follower to brake too hard, avoiding a potential accident.

The incentive criterion is stated as follow:

$$\tilde{a}_c - a_c + p(\tilde{a}_n - a_n) > a_{\text{thr}}, \tag{7}$$

in which $a$ and $\tilde{a}$ denotes the old acceleration and the new acceleration, if the driver decides to change lanes, while the subscripts $c$ and $n$ denotes the current lane-changing driver and its new follower. Note that the left hand side is basically a sum between the advantage of the lane changing driver and the weighted disadvantage of the new follower. The target lane is appealing enough if this sum is bigger than some given threshold $a_{\text{thr}}$. In case the driver has multiple lanes to choose from, the one with the biggest "incentive" sum will be chosen. To enforce asymmetric lane-changing rules, for example "one should keep to the right side of the road", we add more reward when the target lane is the right-side/left-side lane, and punish the driver if it tries to move to the left-side/right-side lane

$$\tilde{a}_c - a_c + p(\tilde{a}_n - a_n) + a_{\text{bias}} > a_{\text{thr}}, \ \text{for a right} - \text{side lane}$$
$$\tilde{a}_c - a_c + p(\tilde{a}_n - a_n) - a_{\text{bias}} > a_{\text{thr}}, \ \text{otherwise} \tag{8}$$

Table 6 describes the parameters that allow to control the lane-change behavior for each drivers.

The model includes additional attributes to control the candidate lanes, i.e. the lanes that will be considered by the MOBIL model. Table 7 presents them.

**Table 6. Attributes related to lane change.**

| Name | Type | Description |
|---|---|---|
| Safe deceleration ($b_{\text{safe}}$) | float | Safe deceleration of the new follower induced by lane changing |
| Politeness factor ($p$) | Float | Attention to others when changing lanes. A high value makes the driver care more about followers' advantages rather than its own |
| Changing threshold ($a_{\text{thr}}$) | Float | Minimum acceleration gain for a driver to change lanes, added to prevent drivers jumping between lanes repeatedly |
| Right/left lane bias ($a_{\text{bias}}$) | Float | Encourages driver to keep to the right/left side of the road |

**Table 7. Selection of candidate lane parameters.**

| Name | Type | Description |
|---|---|---|
| Linked road probability ($p_{\text{linked\_road}}$) | Float | Probability for a driver to consider the linked road (if there is one) |
| Linked road limit ($l_{\text{linked\_road}}$) | Integer | Maximum number of lanes on the linked road that the driver considers to use |
| lane change limit ($l_{\text{limit}}$) | Integer | Maximum "lane range" when switching lanes during segments traversal.[1] |

**Algorithm 2** chooseLane function

```
1: function CHOOSELANE (r)
2:     v^current ← current vehicle
3:     l^best ← current lane
4:     v^lead ← leading vehicle of v^current on l^best of r
5:     a_c ← IDM(v, v^lead)
6:     a_c^best ← a_c
7:     for l ∈ eligible lanes do
8:         v^newLead ← leading vehicle of v^current on l of r
9:         ã_c ← IDM(v^newLead)
10:        v^newBack ← back vehicle of v^current on l of r
11:        v^leadBefore ← leading vehicle of v^newBack on l of r
12:        a_n ← IDM(v^newBack, v^leadBefore)
13:        v^leadAfter ← v^current
14:        ã_n ← IDM(v^newBack, v^leadAfter)
15:        if ã_n <= −b_save then          ▷ safety criterion
16:            continue
17:        end if
18:        inc ← ã_c − a_c + p(ã_n − a_n) + a_bias
19:        if inc > a_thr & inc > inc^best then     ▷ incentive criterion
20:            inc^best ← incentive
21:            l^best ← l
22:            a_c^best ← ã_c
23:        end if
24:    end for
25:    return (l^best, a_c^best)
26: end function
```

## Verification

One important point to be considered when building agent-based models and individual behaviors is their capacity to reproduce known macroscopic behaviors through simulations. In our case, we choose to verify if the use of the model is able to reproduce the shape of the fundamental diagram of a two-phase traffic [38]. The fundamental diagram is based on the following observations:

- when the number of vehicles in the network is very small, interaction between these vehicles is close to zero. Each agent can therefore ride at the maximum speed it wants, without being disturbed by the presence of other vehicles, allowing us to calculate the average maximum speed observed on the network.

- As the number of cars increases, the interaction between vehicles also increases, which has the effect of reducing the speed of the vehicles. As a result, the average speed in the network decreases with respect to the number of vehicles present in the network.

**Table 8. Driver parameter values for the circuit.**

| Name | Value |
| --- | --- |
| Vehicle length (*len*) | $4m$ |
| Max speed ($v_{max}$) | $150 km.h^{-1}$ |
| Speed coefficient | $1.0$ |
| Time headway ($T$) | $1.0s$ |
| Minimum gap ($s_0$) | $0.5m$ |
| Acceleration ($a$) | $2.0 m.s^{-1}$ |
| Deceleration ($b$) | $3.0 m.s^{-1}$ |

- There is a limiting case where the number of vehicles is such that a traffic jam is formed, the average speed is zero and congestion maximum. It is then no longer possible to add vehicles on the section.

In order to test the capacity of the driving skill to reproduce this macroscopic behavior, we carried out a simulation on a simple road infrastructure composed of 4 roads of 2 kilometers connected as a circuit, in which we computed the average speed of the cars. We simulated the traffic, adding progressively more and more vehicles with a speed limit of 50km/h and 1 lane.

Table 8 presents the parameter values used for the driver agents.

We repeatedly: add 100 vehicles; wait for the simulation to stabilize (about 10 minutes); then compute the average speed of the agents over 2 minutes. We go from 0 to 1700 vehicles, as density then makes no traffic possible above that.

Fig 3 shows the average speed according to the number of cars. The chart shows that our model is able to reproduce the different phases of the fundamental diagram as previously described. Indeed, when the number of vehicles is low (less than 400 cars), the average speed of motorists is close to the speed limit of the roads ($50km/h$). Then follows a phase where the speed will decrease more rapidly with the number of cars, until reaching a speed close to 0 representing the complete saturation of the network.

## Computation time

In order to evaluate the capabilities of our model to simulate simultaneously a large number of agents, we measured, using the same methodological framework as before, the computation time taken with different numbers of vehicles.

We performed a simulation on a simple road infrastructure composed of 40 roads of 1.6 kilometers connected in a circuit. We simulated the traffic by progressively adding more and more vehicles with a speed limit of 50km/h and 4 lanes.

After 200 simulation steps (simulation step of 0.5s) used for initialization to avoid artifacts due to the addition of vehicles, we computed the total computation time by step for 100 simulation steps, then we added new vehicles to the circuit. We simulated traffic with 100, 1000, 10000 and 100000 vehicles. We use the same vehicle characteristics as in the previous experiment (Table 8).

Table 9 presents the computation time obtained for different number of vehicles. We have only counted in these computation times the time taken by the movement actions of the vehicles (it does not include the times related to the visualization for example). The simulation was carried out on a Mac mini M1 2022 with 8go of RAM.

The results show that the computation time tends to increase linearly with the number of agents. They also show that the computation time, even with a large number of vehicles, remains reasonable.

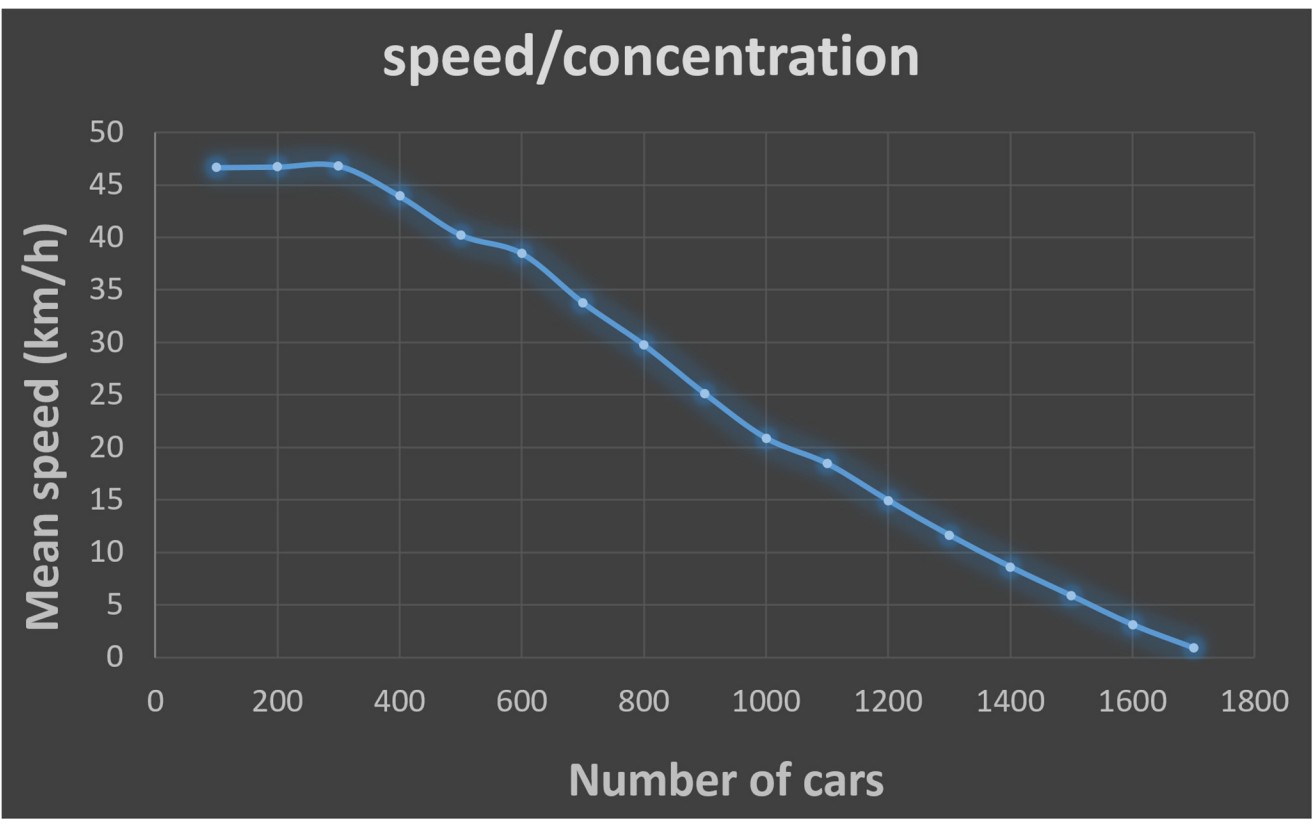

**Fig 3. Average speed value according to the number of cars.**

## 4 Application example: Simulating the traffic of Hanoi

### Context

In order to validate the model, we compare observed and simulated traffic for two sites in Hanoi, Vietnam. Hanoi has undergone great changes in recent years in terms of both the type of mobility, with an increasing presence of cars, and the road infrastructure with the construction of major roads.

Today's road traffic is characterized by an important mix of mobility between motorcycles (the most used means of locomotion) and cars. Indeed, in 2017 [39] conducted a mobility study at Ho Chi Minh City (second-biggest city in Vietnam). The resulting breakdown of travel modes is as follows: 74% of daily travel is done by motorcycle, 19% by bicycle and 1% by car, 4% by bus, and 2% by Taxi. Even if mobility has changed since then, with a consequent increase in the use of cars, this distribution is not so far from the current distribution with a significant predominance of the motorcycle. [40] refers to mixed traffic oriented to motorcycles for this type of traffic.

**Table 9. Computation time according to the number of vehicles.**

| Number of vehicles | Mean computation time per step (in ms) |
| --- | --- |
| 100 | 0.47 |
| 1000 | 4.23 |
| 10 000 | 54.84 |
| 100 000 | 648.89 |

Even though a significant police presence ensures a minimum respect of traffic rules (stop at red light, speed limit), these are still often not respected, in particular by motorcycle drivers. In terms of driving, although there are marked lanes on many roads, they often have little impact on driving habits, especially for motorcycle drivers who do not hesitate to slalom between vehicles: the notion of overtaking on the right or the left does not exist in the practical driving of Hanoians. By convention, cars tend to drive in the left lanes and motorcycles in the right lanes, although this is far from a strict rule. The traffic in Hanoi is often dense and the driving speed rather low.

These characteristics make Hanoi a specific environment for which it can be complex to adapt a traffic simulator.

Nevertheless, various works have already proposed models for the case of Hanoi or for other Vietnamese cities.

In [41], the authors propose a non-lane-based movement model based on the safety distance: according to the location of the other vehicles and to the safety distance, each vehicle computes its acceleration vector. Comparisons with data collected in Ho Chi Minh City showed that the model was able to be reproduced in a realistic way the observed behavior of some motorbikes. If this model is able to reproduce well some behaviors, it is only applicable for simple spatial configuration (straight road without traffic sign). In addition, due to computing performance, it does not allow to scale up and to simulate a large number of vehicles.

A second model to cite is [42]. This work proposes to use a social force model, usually used for pedestrian simulation, to simulate the traffic at intersection. As shown in the article, in its current state, the model's ability to reproduce real traffic is limited due to the lack of attractiveness force.

A third model is [43]. This simple model, developed in Netlogo, proposes to simulate the traffic based on the vision cone of the vehicles. The movement is then dependent on the perceived free spaces. Experiments based on real data collected in Hanoi were used to calibrate the model. However, the paper does not propose any real validation of the model, and, like [41], is limited to simple spatial configuration (straight road) and cannot be used to simulate a large number of vehicles.

A last model to cite is [44]. This model, developed in GAMA, allows to simulate at the scale of a road or a crossroads a mixed traffic (motorcycle, car, bus) in a non-normative context. Several experiments tend to show that the model can reproduce a credible traffic. However, this model, which is based on geometrical calculations, requires precise data on the roads (surface) and is limited in terms of scaling: due to the computation time, it can be only used the simulate a few hundreds of vehicles.

To conclude on this state of the art, there is a real need for a traffic model allowing to take into account the specificity of the traffic in Vietnam while allowing to scale up to simulate the traffic at the scale of a district or a city (with several thousands of vehicles). To verify the ability of our model to simulate traffic in contexts such as Vietnam, we propose to compare the results obtained with our model and the model of [44] with real data collected for two study sites in Hanoi. The following sections describe the data collected, the model built and the results of these comparisons.

## Data collection

To validate the model and thus the ability of GAMA to simulate traffic in Hanoi, we recorded traffic footages on two different roads with different characteristics in terms of number of lanes and density. Two footages were recorded for each of these roads, which corresponds to two different points on the road (input and output). The goal was to have points far enough

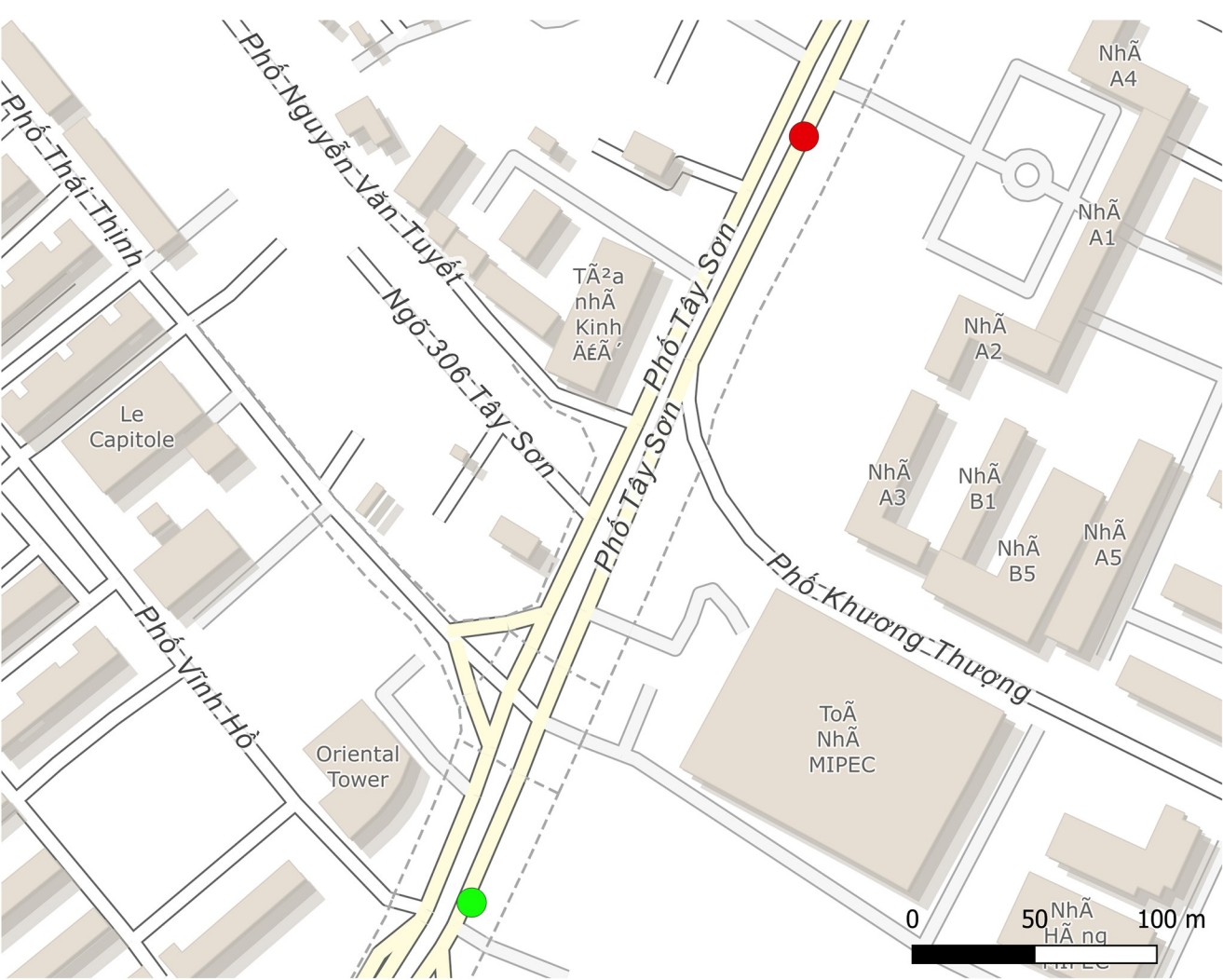

**Fig 4. Tay Son street site (based on OpenStreetMap data): The green circle represents the input point (WGS84 coordinates: (105.8224 21.005)), and the red circle the output (WGS84 coordinates: (105.8236 21.0076)).**

apart to be able to provide more robust data on vehicle travel times and at the same time constitute a main axis so that the vast majority of vehicles passing through point 1 would end up at point 2. Figs 4 and 5 show the sites corresponding to *Tay Son* street and *Chua Boc* street respectively. In both sites, there is a red light in the middle of the road. There is a second red light on the *Chua Boc* street just after the output point that we have integrated into the simulation to take into account the effects of slowing down at the end of the road. In order to give them realistic behaviors, we collected data on the duration of their cycles. Table 10 gives information about the two sites.

In order to count the number of vehicles passing through the input and output points, the footages were passed through a deep learning-based vehicle counter, using Scaled-YOLOv4 [45] for object detection and SORT [46] for tracking. The counter outputs a list of vehicles appearing in a given video, along with their class (vehicle type) and timestamp. In order to convert this list into time series, buckets of one second length were created for each vehicle type. List entries were then distributed into their appropriate bucket based on their

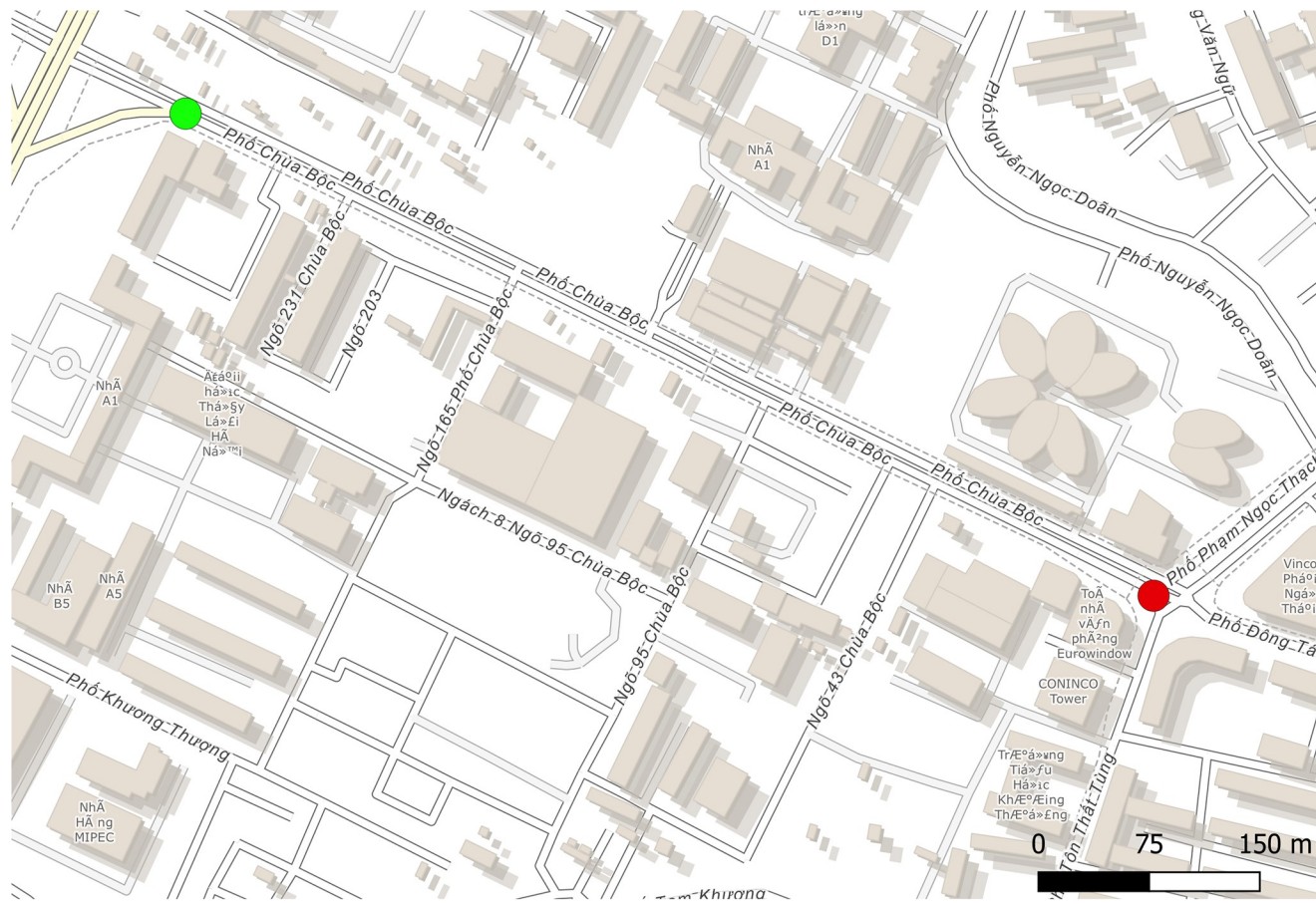

**Fig 5. Chua Boc street site (based on OpenStreetMap data): The green circle represents the input point (WGS84 coordinates: (105.8252 21.0089)), and the red circle the output point (WGS84 coordinates: (105.831 21.0062)).**

timestamps. Figs 6 and 7 present the time series of vehicle count at the input and output points respectively for the Tay Son and Chua Boc sites.

## Description of the model and parameters used

Based on the GAMA integrated model (GAMA 1.8.2), we implemented a simple model in GAML to simulate the traffic flow on the two roads. The number of lanes for each street segment is approximated by dividing the segment width by that of the vehicle type with the

**Table 10. Characteristic of the 2 sites.**

| Name | Tay Son | Chua Boc |
| --- | --- | --- |
| Length | 320m | 667m |
| Number of lanes | 2 | 3 |
| Speed limitation | 40km/h | 40km/h |
| Traffic light cycle (red/green) | 42s/56s | 21s/73s—70s/59s |
| Footage date and duration | July 8, 2021 (Thursday), 10:03am—5 minutes | July 8, 2021 (Thursday), 9:33am—10 minutes |
| Number of vehicles entered during the footage | 1903 | 1031 |

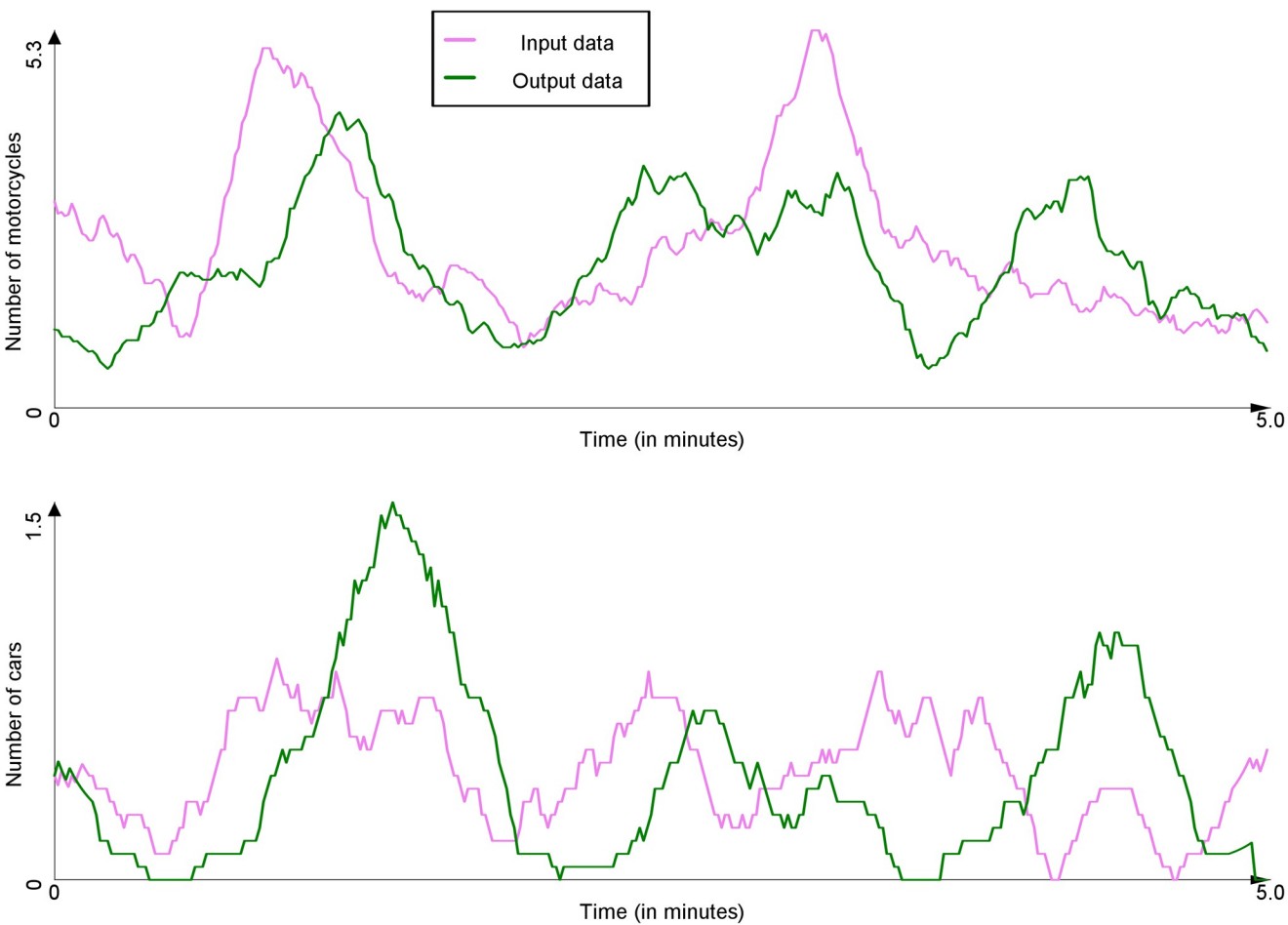

**Fig 6. Count of vehicles at the input and output points for the Tay Son site (time series were smoothed out using moving average with a window size of 10s).**

shortest width. In our case, that would be motorcycles, which we assume to be 0.8-meters wide (typical width is 0.7 meters, and 0.1 meters is for safe spacing). Therefore, when measuring width in terms of number of lanes, a motorcycle takes one lane, while a car takes two. In both scenarios, cars start at the left half of the road, and motorcycles start at the right half, which corresponds to the driving practices in Vietnam. As for the length of the vehicles and their maximum speed, we used values consistent with the vehicle fleet in Hanoi, i.e. respectively 3.8 meters and 2 meters length for cars and motorcycles; $130km/h$ and $60km/h$ for the maximum speed. Note that the maximum speed has little impact in practice: in Hanoi where the traffic is dense, it is very rare for a vehicle to reach its maximum speed. Similarly, by analyzing vehicle behavior at the Tay Son Street, we determined that cars always stop when the light is red while the probability of stopping for motorcycles (in the front row) is only 0.5. Note that the model of [44] does not allow to take into account that vehicles may disregard red lights. In order not to bias the comparison with our model, we have introduced this possibility in the model: a motorcycle arriving at a red light will have only a 50% chance to stop.

The simulation step duration is set to 0.5 seconds.

At the start of the simulation, the time series corresponding to the input point is used to determine when a vehicle would appear and start moving towards the endpoint. Every vehicle

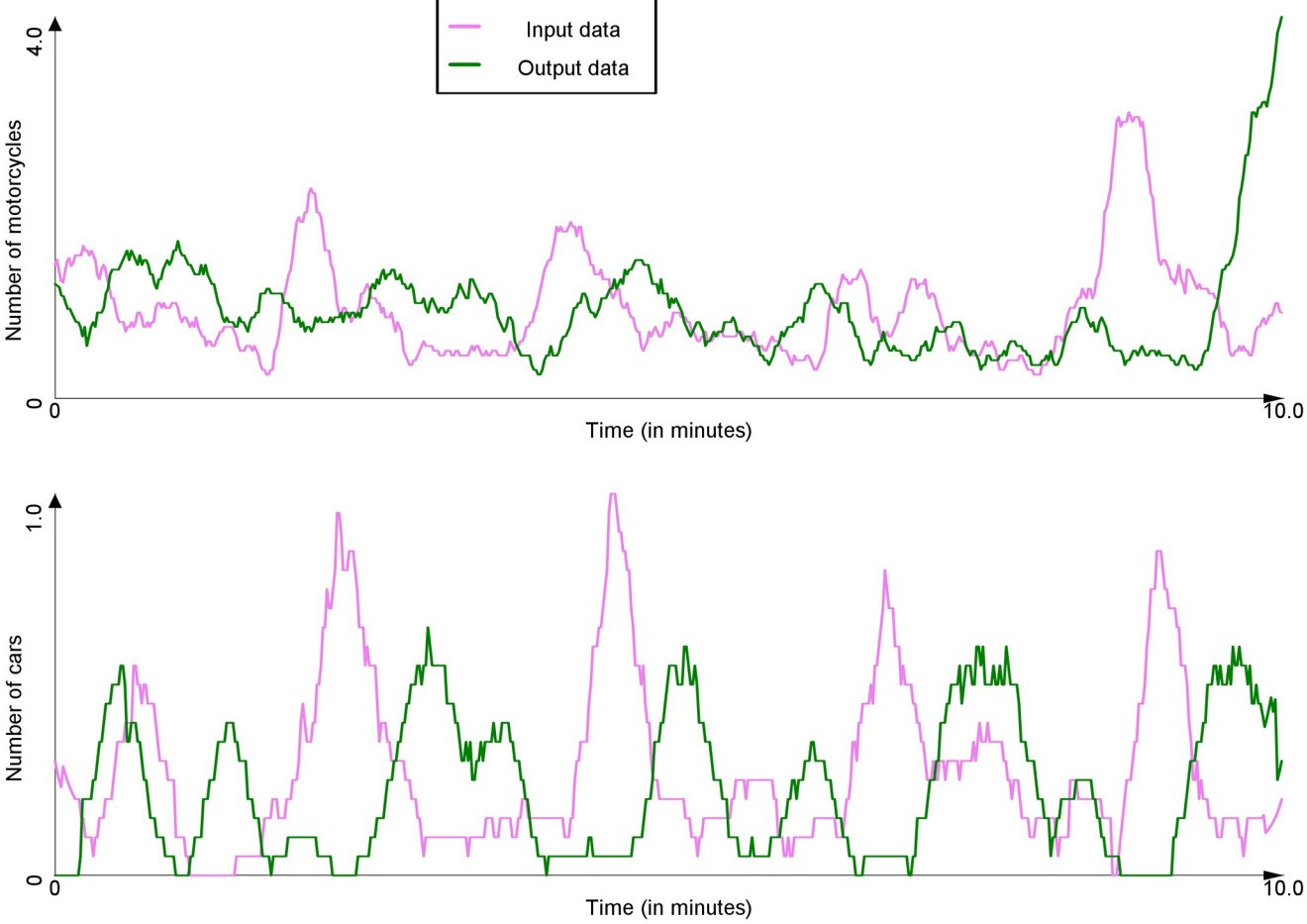

**Fig 7. Count of vehicles at the input and output points for the Chua Boc site (time series were smoothed out using moving average with a window size of 10s).**

start with an initial speed of $20km/h$. Once a vehicle has reached the endpoint, it will be put into the appropriate one-second bucket. The outputs of the simulation are two time series describing the number of vehicles passing the endpoint for motorcycles and cars respectively.

In order to evaluate the difference between the observed and simulated time series we used two classic measures for window size of 10s: sum of squared errors (SSE) and dynamic time warping (DTW) [47].

Regarding model calibration, we used Genetic Algorithms (GA) to find the optimal set of parameters for IDM and MOBIL, i.e. the set which minimizes $DTW_{car} \times N_{car} + DTW_{motorcycle} \times N_{motorcycle}$ for the Tay Son site, with $N_{car}$ and $N_{motorcycle}$ respectively the number of cars and motorcycles in the data. Indeed, data from Tay Son road was used to calibrate the model, while data from Chua Boc was reserved for validation. We proceeded in the same way for the model of [44], for which we calibrated all the parameters in connection with the driving (same algorithm, same fitness function, and same data).

The best parameters found by the calibration process is presented in Table 11.

## Results

For each site, and for the two models ([44] and ours), we replicates 100 times the simulations to ensure the robustness of the simulations.

**Table 11. Driver parameter values for Hanoi.**

| Name | Motorcycle | Car |
|---|---|---|
| Max acceleration ($a$) | $1.1 m.s^{-2}$ | $1.2 m.s^{-2}$ |
| Max deceleration ($b$) | $4.6 m.s^{-2}$ | $4.8 m.s^{-2}$ |
| Time headway ($T$) | $0.8 s$ | $2.5 s$ |
| Safety distance ($s_0$) | $1.9 m$ | $1.1 m$ |
| Politeness factor ($p$) | 0.2 | 0.1 |
| Changing threshold ($a_{thr}$) | 0.3 | 0.15 |
| Safe deceleration ($b_{safe}$) | $1.0 m.s^{-2}$ | $2.5 m.s^{-2}$ |
| Right lane bias ($a_{bias}$) | $0.05 m.s^{-2}$ | $0.45 m.s^{-2}$ |

Figs 8 and 9 presents the number of vehicles count (motorcycles and cars) for the 100 replications for the Tay Son street respectively with the [44] and our model. The red line represents the mean value. These figures already show the moderate impact of stochasticity on the results as well as the observation of cyclical phenomena, which are due to the presence of the traffic light in the middle of the road. Fig 10 presents the comparison between the results obtained with the two models (mean value) and the real data and Table 12 the *SSE* and *DTW* obtained for the motorcycles and cars as well as the computation time (taking only into account the time taken by the vehicle behaviors). A first observation is that our model allows to better reproduce the collected data for cars: both concerning the curve closer to the observed curves, or the values of *SSE* and *DTW* (p-values by Student T test respectively equal to $8.5 \times 10^{-111}$ and $1.5 \times 10^{-3}$), the results are better with our model. For motorcycles, the model of [44] achieves significantly better results in both *SES* and *DTW* (p-values by Student T test

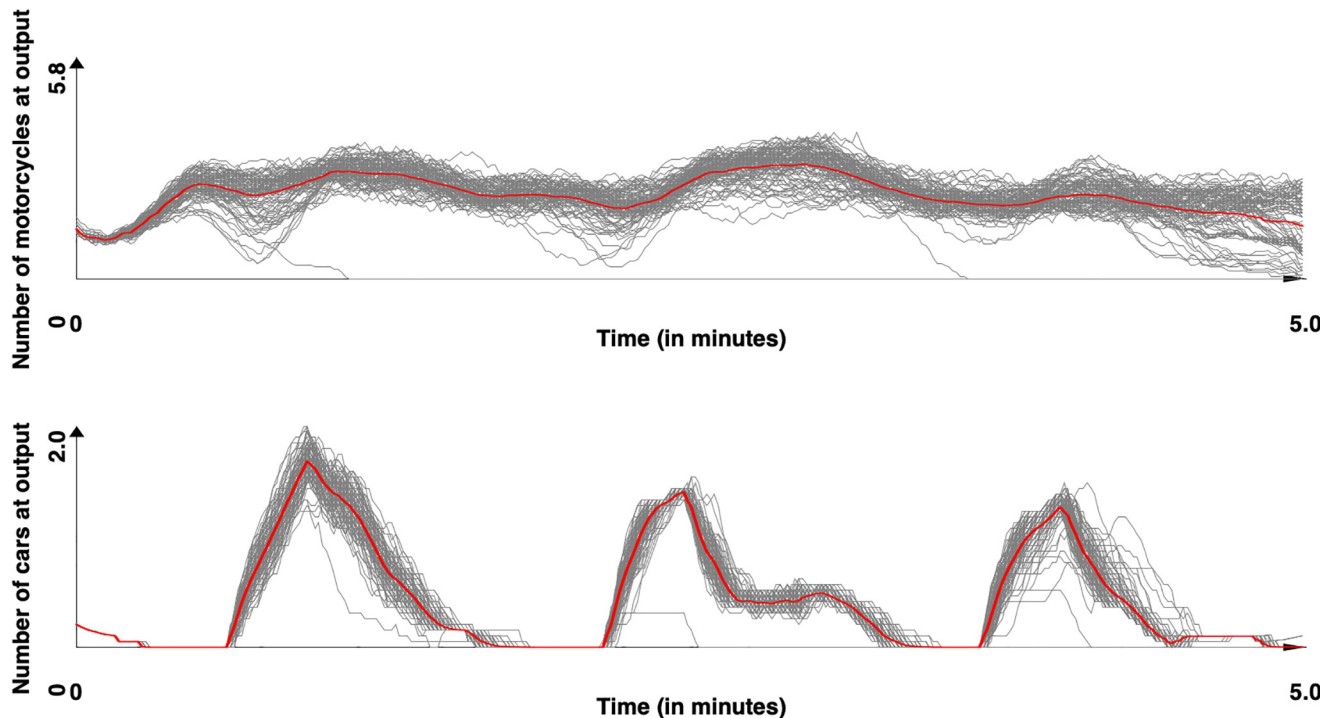

**Fig 8. Simulated results for the Tay Son site.** Motorcycle and car counts for the 100 simulations with [44] model (time series were smoothed out using moving average with a window size of 10s). In red, the mean values.

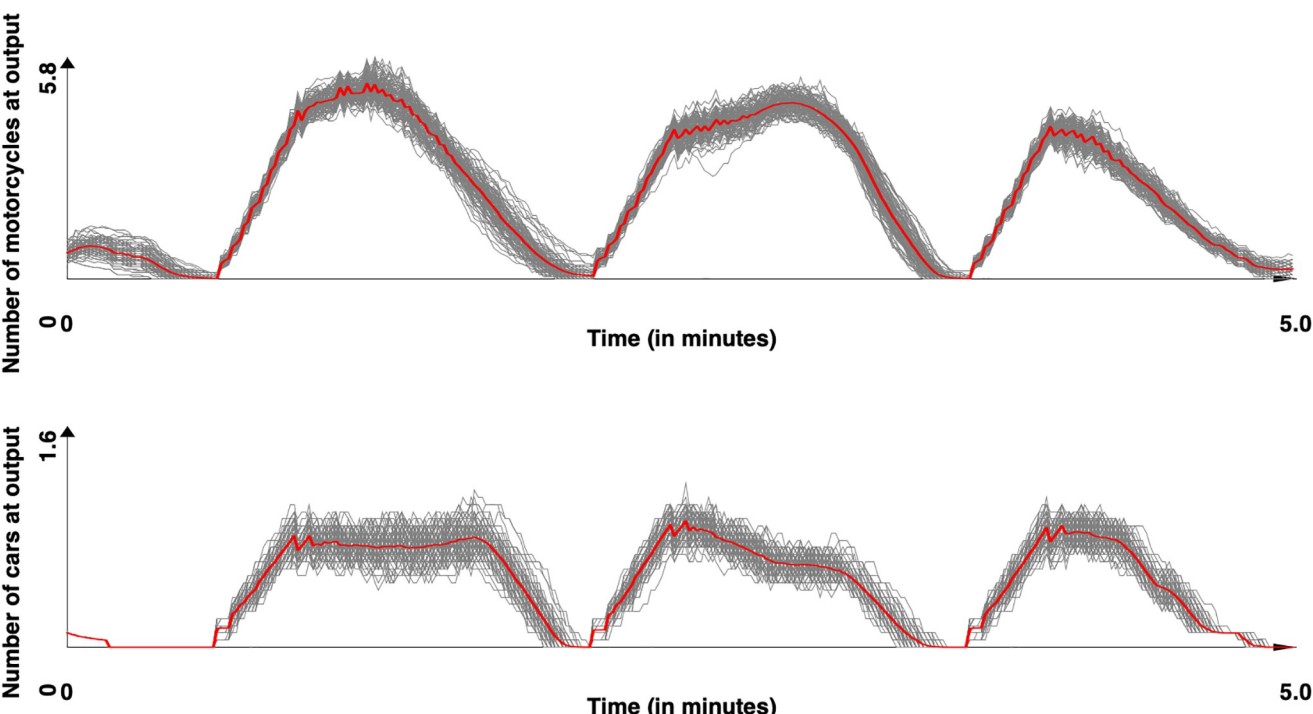

**Fig 9. Simulated results for the Tay Son site.** Motorcycle and car counts for the 100 simulations with our model (time series were smoothed out using moving average with a window size of 10s). In red, the mean values.

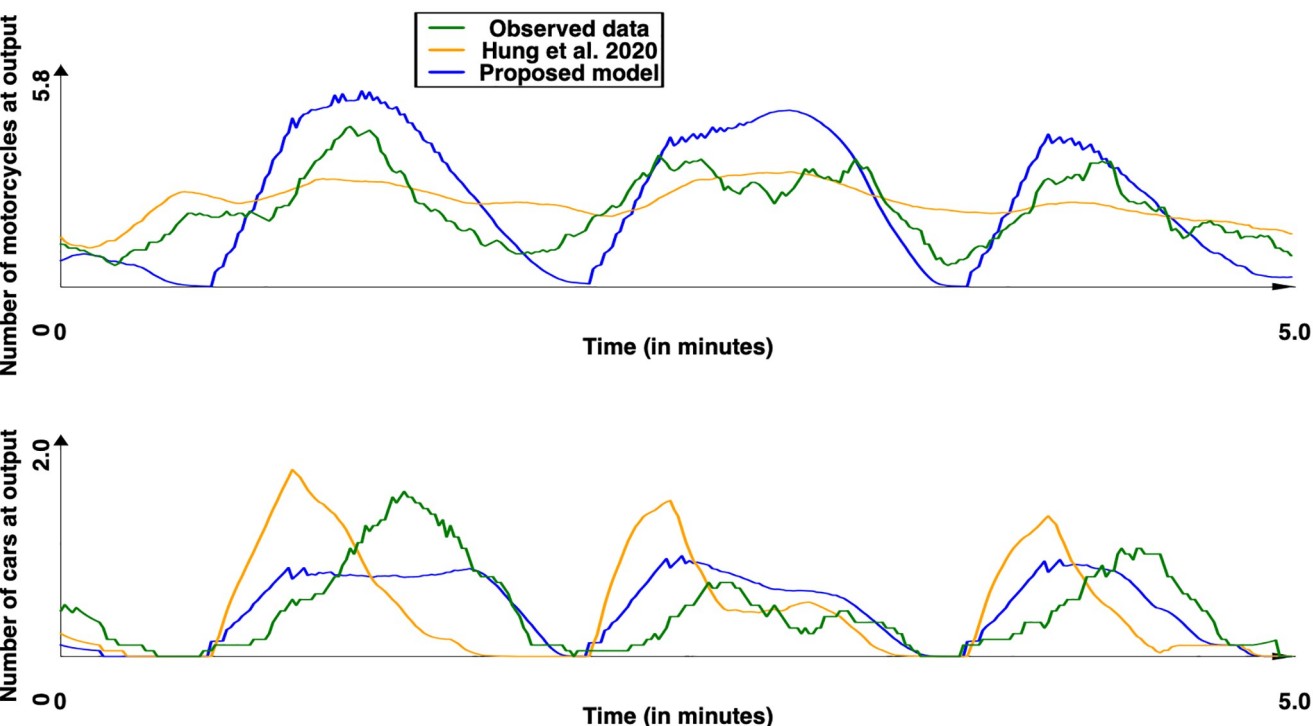

**Fig 10. Observed and simulated results (with [44] and our models) for the Tay Son site.** Motorcycle and car counts—mean of the simulation (time series were smoothed out using moving average with a window size of 10s.

**Table 12. Mean metrics computed for the Tay Son site.** In parenthesis, the standard deviation.

|  | SSE$_{motorcycle}$ | SSE$_{car}$ | DTW$_{motorcycle}$ | DTW$_{car}$ | Computation time (in s) |
|---|---|---|---|---|---|
| [44] | **197.77** (131.24) | 97.43 (8.28) | **106.76** (31.63) | 25.74 (1.98) | 8.14 (14.47) |
| Ours | 354.18 (26.99) | **25.54** (3.12) | 125.15 (10.84) | **24.76** (2.78) | **0.52** (0.07) |

respectively equal to $1.2 \times 10^{-20}$ and $2.4 \times 10^{-7}$). The observation of the curves shows that our model tends to amplify the effects of oscillations compared to the observed data, whereas the model of [44] tends to underestimate them. One explanation is that the vehicles in the model of [43] are not based on lanes but on free spaces to move. Therefore, they have more possibility to execute maneuvers to avoid the vehicles stopped in front of them and therefore, if they decide not to stop at the red light (0.5 probability), continue their way and fluidify the traffic. In our model, the blocking of all lanes at the red light happens more quickly, leaving a more pronounced oscillation phenomenon, closer to what is measured and observed in reality.

We do not find this phenomenon for Chua Boc because the road is much less wide than Tay Son, so even if vehicles have more ability to avoid obstacles, the blocking of spaces at the level of red lights happens much faster and with the phenomenon of oscillation.

Another important element is the time of computation: the simulation is more than 15 times faster with our model, thus allowing to simulate in an acceptable time a much larger number of vehicles.

Concerning the second site, which was not used to calibrate the two models, the results obtained with respectively the models of [44] and our model are presented in Figs 11–13 presents the comparison between the two models and Table 13 the values obtained for *SSE*, *DTW*

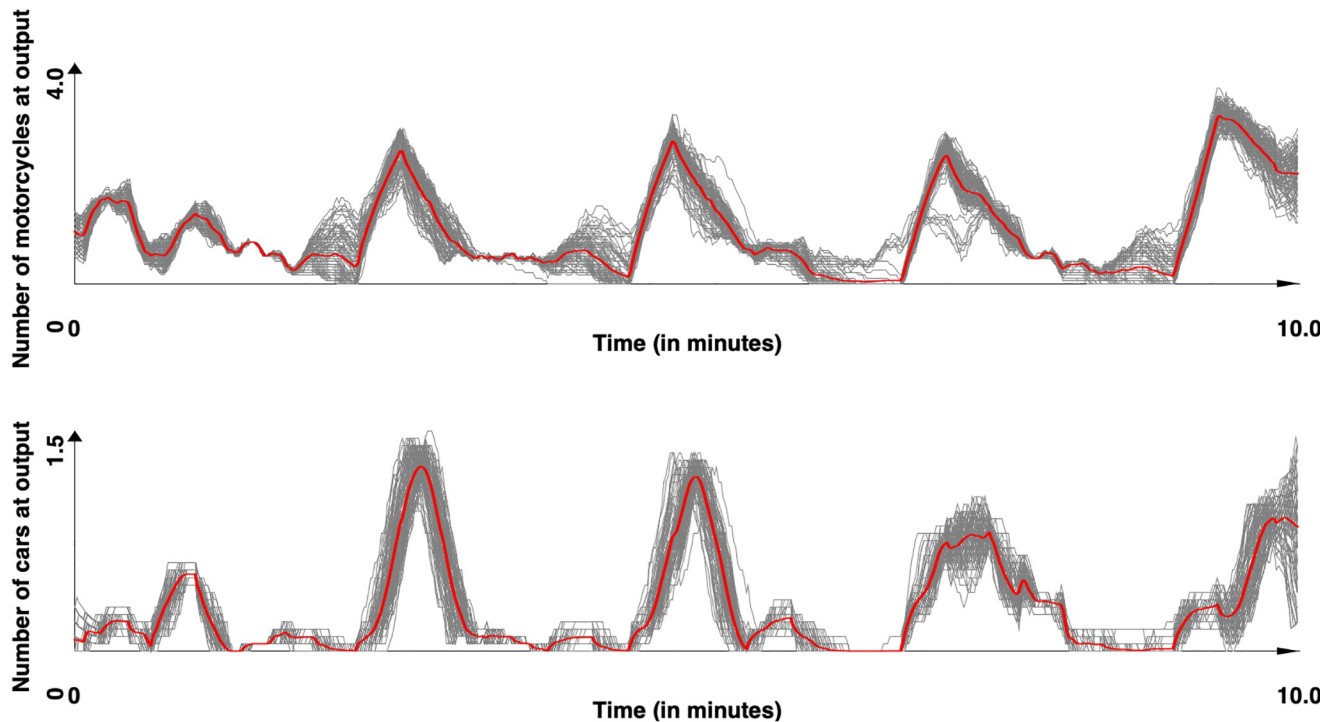

**Fig 11. Simulated results for the Chua Boc site.** Motorcycle and car counts for the 100 simulations with [44] model (time series were smoothed out using moving average with a window size of 10s). In red, the mean values.

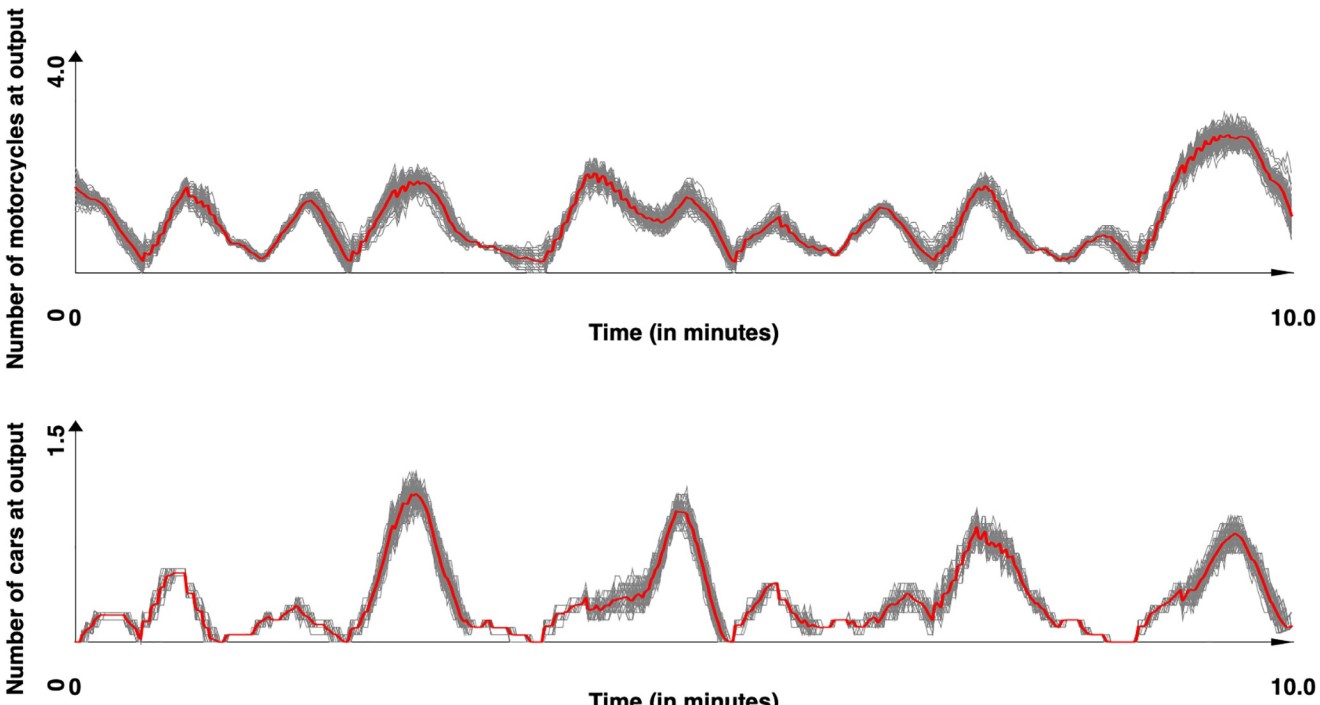

**Fig 12. Simulated results for the Chua Boc site.** Motorcycle and cars counts for the 100 simulations with our model (time series were smoothed out using moving average with a window size of 10s). In red, the mean values.

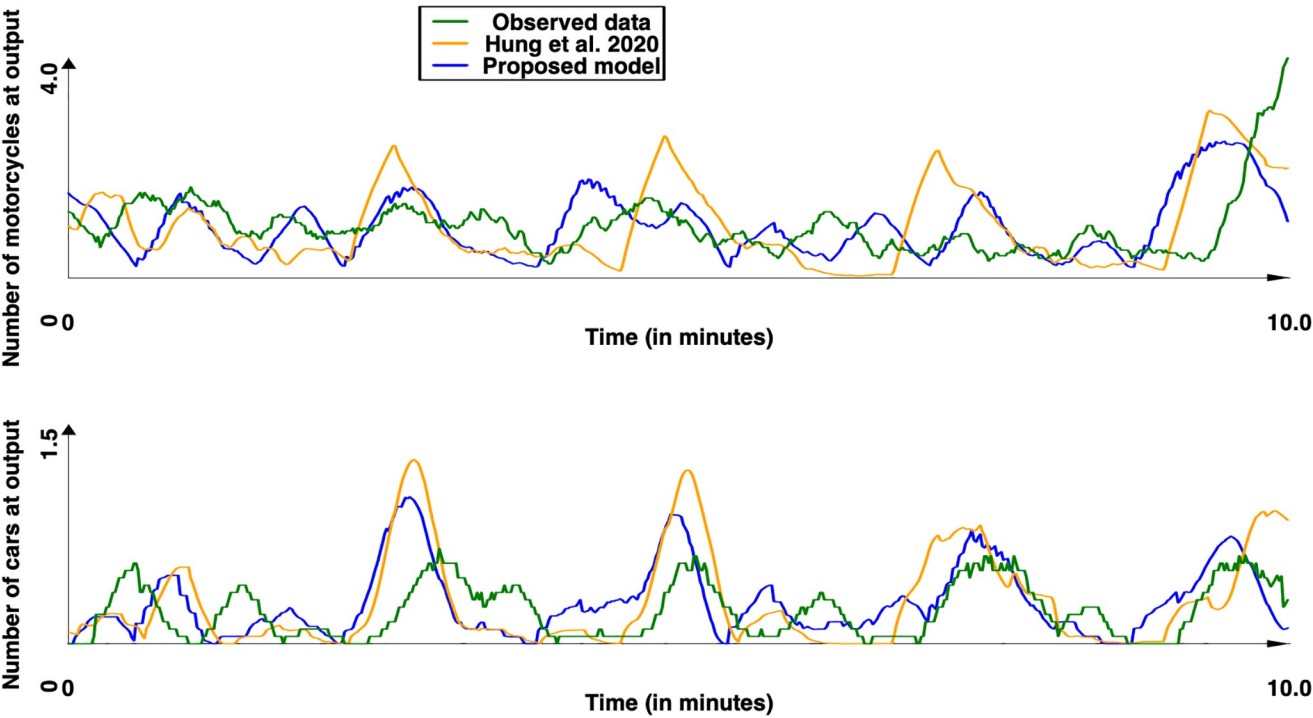

**Fig 13. Observed and simulated results (with [44] and our models) for the Chua Boc site.** Motorcycle and car counts—mean of the simulation (time series were smoothed out using moving average with a window size of 10s.

**Table 13. Mean metrics computed for the Chua Boc site.** In parenthesis, the standard deviation.

|  | SSE$_{motorcycle}$ | SSE$_{car}$ | DTW$_{motorcycle}$ | DTW$_{car}$ | Computation time (in s) |
|---|---|---|---|---|---|
| [44] | 353.5 (34.92) | 48.4 (5.29) | 150.45 (16.46) | 49.39 (3.41) | 5.86 (0.16) |
| Ours | **255.53** (15.68) | **33.5** (1.5) | **100.86** (4.84) | **28.79** (1.49) | **0.54** (0.03) |

and the computation time. For this site, the results show that our model provided results significatively closer to the observed data for both cars and motorcycle (p-values by Student T test lower than $10^{-50}$ for all metrics), with a much lower computation time (over 10 times lower).

## 5 Conclusion

In this paper, we presented a new integrated model for the GAMA platform dedicated to the development of traffic simulations. This integrated model allows to define new traffic simulations with a representation of driver's operational behaviors that is both detailed and easily accessible through the GAML language. In particular, it enables to model road infrastructure and traffic signals, vehicles of different size, change of lanes by the drivers and their tendency to respect norms. We validated the model though an application concerning the simulation of traffic in the city of Hanoi, Vietnam.

Compared to the existing traffic simulation frameworks, the advantage of our tool, in addition to its capacity to simulate mixed and non-normative traffic, is that it allows modelers to easily adapt it to their application context. Indeed, the use of the GAML language enables modelers without high-level programming skills to develop their own models based on the integrate model. For example, this integrated traffic model is now used in the ESCAPE framework [31, 48], which focuses on mass evacuation such as the 270 000 evacuating people simulated in Rouen. The versatility of our integrated model allows Escape to simulate complex mobility behavior (change of plan according to circumstances, non-compliance with traffic rules, etc.), to take into account various types of mobility (car, motorcycle, bus, bicycle, etc.) and to adapt to different countries (France, Vietnam, etc).

The given application case shows that our tool helps to carry out simulations with thousands of driver agents in an acceptable computation time. However, in order to give the possibility of simulating big cities to modelers, we plan to improve the performance of the tool by using distributed computation. Indeed, if the distribution can be complex due to the asynchronous activation of the agents, it may still be possible to distribute a part of the computation burden. Another possibility to allow large scale simulation would be to couple this model with a macro or meso model: the idea, as proposed in [49], is to simulate in detail the roads of interest and the others with less time-consuming model.

Finally, we also plan to develop new tools to help people prepare their data. The goal would be to offer the possibility of automatically filling the missing attributes from incomplete OSM data (OSM are often incomplete), and creating a consistent network (with its infrastructure and traffic signals). Special focus will be on traffic signals and traffic lights.

## Supporting information

**S1 File.**
(PDF)

## Author Contributions

**Conceptualization:** Arnaud Saval, Duc Pham Minh, Kevin Chapuis, Pierrick Tranouez, Clément Caron, Éric Daudé, Patrick Taillandier.

**Data curation:** Duc Pham Minh, Patrick Taillandier.

**Funding acquisition:** Pierrick Tranouez, Éric Daudé.

**Methodology:** Arnaud Saval, Pierrick Tranouez, Patrick Taillandier.

**Project administration:** Éric Daudé.

**Software:** Arnaud Saval, Duc Pham Minh, Kevin Chapuis, Patrick Taillandier.

**Supervision:** Pierrick Tranouez, Patrick Taillandier.

**Validation:** Duc Pham Minh, Kevin Chapuis, Patrick Taillandier.

**Writing – original draft:** Arnaud Saval, Patrick Taillandier.

**Writing – review & editing:** Kevin Chapuis, Pierrick Tranouez, Clément Caron, Éric Daudé.

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
