## [Decision Letter · Decision Letter 0]

28 Mar 2022

PONE-D-22-04270Agent-based Simulation of Traffic with the GAMA Platform: dealing with mixed and non-normative contextsPLOS ONE

Dear Dr. Taillandier,

Thank you for submitting your manuscript to PLOS ONE. After careful consideration, we feel that it has merit but does not fully meet PLOS ONE’s publication criteria as it currently stands. Therefore, we invite you to submit a revised version of the manuscript that addresses the points raised during the review process.

ACADEMIC EDITOR:The reviewers provided comprehensive analysis and suggestions on the paper. Please address. 

We look forward to receiving your revised manuscript.

Kind regards,

Nan Zheng

Academic Editor

PLOS ONE

Journal Requirements:

[This work is funded by the ANR ESCAPE project, grant ANR-16-CE39-0011-01 of the French National Research Agency.]

 [This work is funded by the ANR ESCAPE project, grant ANR-16-CE39-0011-01 of the French National Research Agency (https://anr.fr/)

The funders had no role in study design, data collection and analysis, decision to publish, or preparation of the manuscript.]

4. We note that Figures 4 and 5 in your submission contain satellite images which may be copyrighted. All PLOS content is published under the Creative Commons Attribution License (CC BY 4.0), which means that the manuscript, images, and Supporting Information files will be freely available online, and any third party is permitted to access, download, copy, distribute, and use these materials in any way, even commercially, with proper attribution. For these reasons, we cannot publish previously copyrighted maps or satellite images created using proprietary data, such as Google software (Google Maps, Street View, and Earth). For more information, see our copyright guidelines: http://journals.plos.org/plosone/s/licenses-and-copyright.

a) You may seek permission from the original copyright holder of Figures 4 and 5 to publish the content specifically under the CC BY 4.0 license.  

Additional Editor Comments:

The reviewers provided comprehensive analysis and suggestions for the paper. Please address.

Reviewers' comments:

Reviewer's Responses to Questions

**Comments to the Author**

1. Is the manuscript technically sound, and do the data support the conclusions?

Reviewer #1: Yes

Reviewer #2: Partly

Reviewer #3: Yes

2. Has the statistical analysis been performed appropriately and rigorously? 

Reviewer #1: No

Reviewer #2: Yes

Reviewer #3: N/A

3. Have the authors made all data underlying the findings in their manuscript fully available?

Reviewer #1: No

Reviewer #2: Yes

Reviewer #3: Yes

4. Is the manuscript presented in an intelligible fashion and written in standard English?

Reviewer #1: Yes

Reviewer #2: Yes

Reviewer #3: Yes

5. Review Comments to the Author

Reviewer #1: There are some elements within the writing style which might be considered too informal for a journal paper - for example the first sentence "These last few years have seen the number of traffic based frameworks..." Should read something like "Recent developments [cite???] have resulted in an increase in the number of traffic based frameworks....". I would suggest, as a general measure, that you have your paper proof-read and issues such as this dealt with. But to balance the above comments, the text reads well and communicates the thoughts and ideas of the author to readers (which is of course the prime reason for writing a paper!).

The authors might want to check that they have included ALL relevant previous work (e.g. https://www.springerprofessional.de/en/using-agade-traffic-to-analyse-purpose-driven-travel-behaviour/19696732 and https://link.springer.com/chapter/10.1007/978-3-319-59930-4_37

An area that is not addressed by the authors when discussing the various platforms and technologies for ABM of traffic the means of accessing the simulation. For example, many of these simulation frameworks have to be downloaded, installed and are used in conjunction with programming languages such as C++ or Java. This requires considerable computer science skills, which might not be possessed by a transportation analyst. It may be appropriate to consider the ease of use from the perspective of a transportation specialist when evaluating them.

Algorithm 1 is split across 2 pages!

Please be consistent in the style and notation adopted within your algorithmic code.

Results comprise of simulations covering two sites. A basic comparison is made with [43] and an improvement is noted. the results could be improved by giving some indication as to whether the improvement can be considered statistically significant or not? The lack of a statistical analysis is a major omission from a paper that is otherwise sound.

Reviewer #2: This paper deals with a very interesting topic of simulating non-normative traffic conditions. The paper is well structured and easy to follow.

Abstract

Try to avoid ellipsis points in a scientific text.

As you are talking about microscopic traffic simulations I would leave MATSim out of the discussion, as MATSim is mesoscopic and does not aim to model traffic in such detail.

You say that this new model can simulate tens of thousands of agents, but this is not shown in the paper. Moreover, urban areas today have millions of people. Can you deal with such a scale?

Introduction

Lines 65-66, why is it advantageous that your model is integrated with GAMA? Why is, for instance, SUMO not easily adaptable to new contexts while your approach is? Can you please provide some concrete examples?

Related works

I think that the MATSim community prefers a different citation:

Horni, A., K. Nagel and K. W. Axhausen (2016) The multi-agent transport simulation MATSim, Ubiquity Press London.

Sometimes you write JAVA and sometimes Java, I believe Java would be more appropriate.

It is unclear what is meant by “Synthetic population generation is highly dependent on these modules…”. I would argue that MATSim is independent of the process used to generate the synthetic population. Please elaborate.

SUMO community also prefers a different citation (as stated on the website):

Pablo Alvarez Lopez, Michael Behrisch, Laura Bieker-Walz, Jakob Erdmann, Yun-Pang Flötteröd, Robert Hilbrich, Leonhard Lücken, Johannes Rummel, Peter Wagner, and Evamarie Wießner. IEEE Intelligent Transportation Systems Conference (ITSC), 2018.

Line 117, you mention advanced features, please give examples.

The GAMA integrated traffic model

Lines 207-210, the sentence is badly written and needs re-writing.

Line 226, occupied-> occupies

Line 234 find->finds

Line 273, 1 -> Algorithm 1

Line 274, feature -> features

Ine 307, vanishes -> vanish

Why is the incentive criterion (equation 7) containing the disadvantage for the new follower? The one changing the lane usually only looks at her own advantage. Can you please explain why did you consider also this second term in the equation?

Verification

You should make it clear that you are not validating your model using an empirical MFD, but only that the shape of the graph resembles an expected MFD shape.

I assume you are adding 100 vehicles to the system every 2 minutes? This is not clear from lines 394-396.

Application example

The context could go to the related works section.

Line 412, please state i which year.

Lines 415-417, if mobility -> even if mobility

Data collection

Line 478, a list a -> a list of

Description of the model and parameters used

Line 491, two lane -> two lanes

2 meters is much more than double of 0.8 meters.

How many vehicles are simulated in the presented results? I guess much less then thousands you were mentioning before.

Results

Can you please explain how is the model from [43] calibrated? This is not even explained in the original manuscript as far as I can see. The article [43] is, I guess, not allowing agents to violate red lights at intersections (as can be seen from figures 10 and 13). I wonder if this simple addition is added how would the two models compare in terms of DTW.

It is not clear to me if the road in Figure 4 has another approach at the intersection. The figure is of bad quality, as in general, all figures are ( please improve the quality/resolution of figures). For the roads, a schematic would have been nicer and a pointer to the WGS84 coordinates of start and end points, so one could easily find them.

I also wonder how your model compares with SUMO as a notable example of an open-source traffic simulation. How much effort would it be to add a probability function in SUMO that allows agents to violate red signals at intersections? I wonder how much of the added value is exactly this feature compared with the non-normative driving behavior before reaching the signal. This you could even test using your own model, and by switching on and off different features.

Please try to avoid using etc. in scientific texts.

Reviewer #3: The paper provides a modeling framework for representing non-normative travel behavior among drivers in multimodal traffic. The focus is on how to model road infrastructure, traffic signals and motocycles in a non-normative scenario where rules of traffic are not standardized or necessarily obeyed. The authors do this by integrating their model in GAMA an open source generic agent based simulation platform. Their model is written in GAML and this has been done in order to make their model easily accessible.

The major contribution of this work is the authors' demonstration of how to model non-normative contexts that do not follow international conventions or general traffic rules, such as, drivers changing lanes or overtaking other drivers while driving against traffic flow, disobeying traffic lights, etc. The model serves for small scale scenarios although it would be good to know how it performs for large scale scenarios.

Below are my comments and suggestions

It would be good to provide a reference to back up this statement on Line 21 - 24.

Line 64 - 67 suggest that other exisitng modeling frameworks do not take account of multi-modal traffic which is not so. Unless the authors mean to say that these frameworks do not account specifically for multi-modal trafic in non-normative context. In that case the statement needs to be adjusted to say that.

In Line 85 the authors wrongly states that VISSIM, AnyLogic etc are agent based modeling frameworks. The author should correct this considering that in the citation of Nguyen et al [16] they review the agent-based models. Also it is not clear why TRANSIMS has been reviewed for comparison.

In Line 101 to 103 the author mentions that MATSim do not have extensions with non normative driver behavior and mixed interacting modes. However it should be mentioned that MATSim is a mesoscopic traffic simulator which is why it does not do this.

Authors should specify the meanings of the abbreviations for IDM and MOBIL when first written.

In section 4.1, it is not clear what the blue dot on Figure 1 represents. Is it a node connecting different sections of the roads? Also the concept of sublanes is not clear. It would be nice to illustrate the sublanes in Figure 2. For example in Line 226 to 228, is the car meant to occupy 2 sublanes or 2 lanes?

In equation 1, using s to represent distance could be confusing.

It is not clearly explained how the vehicle speeds are computed in the model even though this is shown in the equations.

Is equation 3 not referring to the computation of vehicle freespeed v_not?

For section 5, the GAMA version used for the scenario testing should be specified. The year of the mobility study used for mode share reference should be specified as well

What the authors want to say in Line 415 - 417 is not immediately clear

For which city is the work by [41, 43]? Stating it would give better context to the comparisons. A table summarising these comparisons would also be apt.

In the description of their model for Hanoi the author should specify if peak traffic period or the whole day is modelled. It should also be justified why an initial speed of 20km/h has been given to every vehicle at the start of the simulation. Is it based on equation 3 or just random?

What does cycling phenomena refer to in line 521?

General suggestions

It would be nice to have a general figure of how the authors' model integrates with the the GAMA simulation

Check for grammatical errors e.g:

Line 207, depending of the countries and drivers, a lane can be compare as a corridor  depending on the countries and drivers, a lane can be compared to a corridor...

line 302 driver to drives  driver to drive

Line 419 If a significant  Even though a significant...

Line 434 was able to reproduced  was able to reproduce

Line 454 it can be only used the simulate only hundreds  it can only be used to simulate hundreds...

Check for future tense wrong usage. E.g in Table 2 the purpose will be covered Page 6  the purpose is covered in page 6

Check for references to the algorithms. See line 273 and 283

Maybe revise the title to capture the essence of the paper. E.g. Dealing with mixed and non-normative traffic. An agent-based simulation with the GAMA platform.

6. PLOS authors have the option to publish the peer review history of their article (what does this mean?). If published, this will include your full peer review and any attached files.

Reviewer #1: No

Reviewer #2: No

Reviewer #3: No

---

## [Author Response · Author response to Decision Letter 0]

21 Jul 2022

RESPONSE TO REVIEWERS

ACADEMIC EDITOR:

The reviewers provided comprehensive analysis and suggestions on the paper. Please address. 

A: Protocols.io seems better adapted to life and matter sciences. For our computer science model, we open sourced the code and the relevant data to allow for reproducibility.

4. We note that Figures 4 and 5 in your submission contain satellite images which may be copyrighted. (...)

A: We completely changed these figures, and replaced the images with OpenStreetMaps maps.

REVIEWER #1:

There are some elements within the writing style which might be considered too informal for a journal paper - for example the first sentence "These last few years have seen the number of traffic based frameworks..." Should read something like "Recent developments [cite???] have resulted in an increase in the number of traffic based frameworks....".

A: Rewritten.

I would suggest, as a general measure, that you have your paper proof-read and issues such as this dealt with. But to balance the above comments, the text reads well and communicates the thoughts and ideas of the author to readers (which is of course the prime reason for writing a paper!).

A: Although you mentioned it was legible, we tried to improve it further.

The authors might want to check that they have included ALL relevant previous work (e.g. https://www.springerprofessional.de/en/using-agade-traffic-to-analyse-purpose-driven-travel-behaviour/19696732 and https://link.springer.com/chapter/10.1007/978-3-319-59930-4_37

A: We have not included ALL previous works on the subject, as it is too vast an endeavor, only the most important papers and frameworks. Agade seems to be interesting, but it is not easy to be sure as the paper quoted above which seems to describe the whole system is a 4-page conference paper with 1 citation. We did extensively quote the great review of agent-based traffic frameworks by the same team. 

An area that is not addressed by the authors when discussing the various platforms and technologies for ABM of traffic the means of accessing the simulation. For example, many of these simulation frameworks have to be downloaded, installed and are used in conjunction with programming languages such as C++ or Java. This requires considerable computer science skills, which might not be possessed by a transportation analyst. It may be appropriate to consider the ease of use from the perspective of a transportation specialist when evaluating them.

A: This is an excellent remark and we tried to explain this at the end of our introduction.

Please be consistent in the style and notation adopted within your algorithmic code.

A: Improved

Results comprise of simulations covering two sites. A basic comparison is made with [43] and an improvement is noted. the results could be improved by giving some indication as to whether the improvement can be considered statistically significant or not? The lack of a statistical analysis is a major omission from a paper that is otherwise sound.

A: Absolutely true. We performed the analysis and included it in the text.

REVIEWER #2:

This paper deals with a very interesting topic of simulating non-normative traffic conditions. The paper is well structured and easy to follow.

Try to avoid ellipsis points in a scientific text.

A: Rewritten.

As you are talking about microscopic traffic simulations I would leave MATSim out of the discussion, as MATSim is mesoscopic and does not aim to model traffic in such detail.

A: We incorporated in our explanation that part of the reason why MATSim is not capable of handling mixed and non-normative traffic is that it works mainly at a mesoscopic scale, nonetheless it seems to us that it is too well-known a traffic framework for us not to mention it altogether.

You say that this new model can simulate tens of thousands of agents, but this is not shown in the paper. Moreover, urban areas today have millions of people. Can you deal with such a scale?

A: True, thank you. We performed and included in the paper a new experiment where we simulate up to 100 000 agents in the traffic.

Lines 65-66, why is it advantageous that your model is integrated with GAMA? Why is, for instance, SUMO not easily adaptable to new contexts while your approach is? Can you please provide some concrete examples?

A: GAMA is a generic platform that is already used to build models in very different domains (opinion dynamics, epidemiology, natural resource management, risk management, transportation…). Many modelers using GAMA do not come from a computer science background and would have great difficulty programming their model with a generic language like C++. Thus, one of the big advantages of integrating the model directly into GAMA and not just trying to enrich SUMO is to facilitate the interaction of our traffic model with other models: the fact of staying in the same platform handling the same objects and based on the same abstraction is much easier than coupling different software. For example, just adding new elements such as vehicle-pedestrian interaction inside SUMO is complex whereas it can be done very simply with GAMA (which already integrated some classic built-in pedestrian models). We tried to better explain this aspect in the manuscript.

I think that the MATSim community prefers a different citation:

Horni, A., K. Nagel and K. W. Axhausen (2016) The multi-agent transport simulation MATSim, Ubiquity Press London.

A: We changed the citation.

Sometimes you write JAVA and sometimes Java, I believe Java would be more appropriate.

A: Java is now used everywhere.

It is unclear what is meant by “Synthetic population generation is highly dependent on these modules…”. I would argue that MATSim is independent of the process used to generate the synthetic population. Please elaborate.

A: It was indeed not very clear nor did it contribute much: we took it out

SUMO community also prefers a different citation (as stated on the website):

Pablo Alvarez Lopez, Michael Behrisch, Laura Bieker-Walz, Jakob Erdmann, Yun-Pang Flötteröd, Robert Hilbrich, Leonhard Lücken, Johannes Rummel, Peter Wagner, and Evamarie Wießner. IEEE Intelligent Transportation Systems Conference (ITSC), 2018.

A: We changed the citation.

Line 117, you mention advanced features, please give examples.

A: We took out “advanced” if it was the cause of your question. 

The GAMA integrated traffic model

Lines 207-210, the sentence is badly written and needs re-writing.

Line 226, occupied-> occupies

Line 234 find->finds

Line 273, 1 -> Algorithm 1

Line 274, feature -> features

Ine 307, vanishes -> vanish

A: Re-written and corrected.

Why is the incentive criterion (equation 7) containing the disadvantage for the new follower? The one changing the lane usually only looks at her own advantage. Can you please explain why did you consider also this second term in the equation?

A: We used the equation as in the original MOBIL paper, to which you can refer for further details. 

You should make it clear that you are not validating your model using an empirical MFD, but only that the shape of the graph resembles an expected MFD shape.

A: Done.

I assume you are adding 100 vehicles to the system every 2 minutes? This is not clear from lines 394-396.

A: Yes, that's what we do. Rewritten for improved clarity

The context could go to the related works section.

A: It is the context of our experiment, so we do not think so.

Line 412, please state i which year.

Lines 415-417, if mobility -> even if mobility

A: Done.

Line 478, a list a -> a list of

A: Done.

Line 491, two lane -> two lanes

A: Done.

2 meters is much more than double of 0.8 meters.

A: Rewritten

How many vehicles are simulated in the presented results? I guess much less then thousands you were mentioning before.

A: Number added in the text (1903 and 1031)

Can you please explain how is the model from [43] calibrated? This is not even explained in the original manuscript as far as I can see. The article [43] is, I guess, not allowing agents to violate red lights at intersections (as can be seen from figures 10 and 13). I wonder if this simple addition is added how would the two models compare in terms of DTW.

A: Excellent remark, thank you. We redid the experiments described in [43] while adding the possibility to violate red lights, and wrote our calibration and comparison in the paper.

It is not clear to me if the road in Figure 4 has another approach at the intersection. The figure is of bad quality, as in general, all figures are ( please improve the quality/resolution of figures). For the roads, a schematic would have been nicer and a pointer to the WGS84 coordinates of start and end points, so one could easily find them.

A: We redrew all the figures.

I also wonder how your model compares with SUMO as a notable example of an open-source traffic simulation. How much effort would it be to add a probability function in SUMO that allows agents to violate red signals at intersections? I wonder how much of the added value is exactly this feature compared with the non-normative driving behavior before reaching the signal. This you could even test using your own model, and by switching on and off different features.

A: It is indeed possible to integrate in SUMO some of the elements that we have integrated in our model (in particular on the non-normative aspects of driving) to allow it to better simulate road behaviors as they can be observed in Vietnam. Nevertheless, it seems interesting to us to integrate our model in GAMA rather than in a platform like SUMO. We have tried to argue this point better in the introduction. For the two roads we studied, it is certain that allowing motorcycles not to stop at a red light is an important element that strongly impacts the result (and that is why, following your suggestion, we added this feature to the Dang et al. model for comparison).

Please try to avoid using etc. in scientific texts.

A: Done.

REVIEWER #3:

It would be good to provide a reference to back up this statement on Line 21 - 24.

A: We could not find a paper where equational macroscopic models model individual behavior such as driving against the flow or with multiple vehicles sharing the width of a lane. We could not either find a reference saying that such a thing does not exist. 

Line 64 - 67 suggest that other exisitng modeling frameworks do not take account of multi-modal traffic which is not so. Unless the authors mean to say that these frameworks do not account specifically for multi-modal trafic in non-normative context. In that case the statement needs to be adjusted to say that.

A: Adjusted.

In Line 85 the authors wrongly states that VISSIM, AnyLogic etc are agent based modeling frameworks. The author should correct this considering that in the citation of Nguyen et al [16] they review the agent-based models. Also it is not clear why TRANSIMS has been reviewed for comparison.

A: They are well-known microsimulation capable traffic modeling softwares or framework, that we only mention to explain why, although they may sound familiar to the reader, especially if he or she comes from a traffic simulation culture, they are precisely out of the scope of this short survey.

In Line 101 to 103 the author mentions that MATSim do not have extensions with non normative driver behavior and mixed interacting modes. However it should be mentioned that MATSim is a mesoscopic traffic simulator which is why it does not do this.

A: We added this mention

Authors should specify the meanings of the abbreviations for IDM and MOBIL when first written.

A: Done.

In section 4.1, it is not clear what the blue dot on Figure 1 represents. Is it a node connecting different sections of the roads? Also the concept of sublanes is not clear. It would be nice to illustrate the sublanes in Figure 2. For example in Line 226 to 228, is the car meant to occupy 2 sublanes or 2 lanes?

A: The blue node is indeed a node connecting two sections of the road, while not being an intersection. It can be used to model geometry or a change in some characteristics of the section (number of lanes, speed limits) 

In equation 1, using s to represent distance could be confusing.

A: It is the traditional name of this variable, including in [9] where the IDM is defined.

It is not clearly explained how the vehicle speeds are computed in the model even though this is shown in the equations.

A: A more intuitive explanation can be found in [9] and its references. The operational principle was that we started with a well-established car-following model in freeways, that we adapted to urban traffic in an agent-based simulation.

Is equation 3 not referring to the computation of vehicle freespeed v_not?

A: It was erroneously mentioning S0 on the left side of the equation, we replaced it by the correct v0.

For section 5, the GAMA version used for the scenario testing should be specified. The year of the mobility study used for mode share reference should be specified as well

A: Done.

What the authors want to say in Line 415 - 417 is not immediately clear

A: Re-written.

For which city is the work by [41, 43]? Stating it would give better context to the comparisons. A table summarising these comparisons would also be apt.

A: Vietnamese cities, Hanoi and Ho Chi Minh city. We indicated it in the text.

In the description of their model for Hanoi the author should specify if peak traffic period or the whole day is modelled. It should also be justified why an initial speed of 20km/h has been given to every vehicle at the start of the simulation. Is it based on equation 3 or just random?

A: Some precisions have been added in the text.

What does cycling phenomena refer to in line 521?

A: Cyclical, not cycling, fixed.

It would be nice to have a general figure of how the authors' model integrates with the the GAMA simulation

A: It is a plugin integrated as an agent's skill, we could not find a figure to make this clearer. We rather added a sentence to clarify this in the introduction.

Line 207, depending of the countries and drivers, a lane can be compare as a corridor  depending on the countries and drivers, a lane can be compared to a corridor...

line 302 driver to drives  driver to drive

Line 419 If a significant  Even though a significant...

Line 434 was able to reproduced  was able to reproduce

Line 454 it can be only used the simulate only hundreds  it can only be used to simulate hundreds...

Check for future tense wrong usage. E.g in Table 2 the purpose will be covered Page 6  the purpose is covered in page 6

A: Done and corrected.

Check for references to the algorithms. See line 273 and 283

Maybe revise the title to capture the essence of the paper. E.g. Dealing with mixed and non-normative traffic. An agent-based simulation with the GAMA platform.

A: Great suggestion, done.

---

## [Decision Letter · Decision Letter 1]

30 Aug 2022

PONE-D-22-04270R1Dealing with mixed and non-normative traffic. An agent-based simulation with the GAMA platformPLOS ONE

Dear Dr. Taillandier,

Thank you for submitting your manuscript to PLOS ONE. After careful consideration, we feel that it has merit but does not fully meet PLOS ONE’s publication criteria as it currently stands. Therefore, we invite you to submit a revised version of the manuscript that addresses the points raised during the review process.

Please address the remaining comment. It looks like the paper is ready soon.==============================

We look forward to receiving your revised manuscript.

Kind regards,

Nan Zheng

Academic Editor

PLOS ONE

Journal Requirements:

Reviewers' comments:

Reviewer's Responses to Questions

**Comments to the Author**

1. If the authors have adequately addressed your comments raised in a previous round of review and you feel that this manuscript is now acceptable for publication, you may indicate that here to bypass the “Comments to the Author” section, enter your conflict of interest statement in the “Confidential to Editor” section, and submit your "Accept" recommendation.

Reviewer #2: All comments have been addressed

Reviewer #3: All comments have been addressed

2. Is the manuscript technically sound, and do the data support the conclusions?

Reviewer #2: Yes

Reviewer #3: Yes

3. Has the statistical analysis been performed appropriately and rigorously? 

Reviewer #2: Yes

Reviewer #3: N/A

4. Have the authors made all data underlying the findings in their manuscript fully available?

Reviewer #2: Yes

Reviewer #3: Yes

5. Is the manuscript presented in an intelligible fashion and written in standard English?

Reviewer #2: Yes

Reviewer #3: Yes

6. Review Comments to the Author

Reviewer #2: I have no further comments.

Reviewer #3: Thank you for responding to the revisions suggested. The response to the suggested revisions is satisfactory and the paper has been improved for more clarity. However, the authors need to pay careful attention to the following and revise the manuscript carefully before publishing can be done:

- Incomplete sentences e.g., line 333 page 10 "(Function :"

- Poorly formatted references in the text and poor latex formats e.g., line 300 page 10 "textbfchooseLane" and line 306 page 10 "see ??"

- Grammatical mistakes e.g line 117 page 4 e.g. having a bicycle or not

- Others

- Line 177 page 5, it is not clear if authors refer to the Table 1 in the article being referenced

- Need for consistency. Line 265 page 8. Compute_path written differently from other Java methods highlighted

These are just a few examples. Further proofreading is necessary for the whole manuscript.

Additionally, for the results it would be beneficial to readers if the authors include a brief explanation on the reason behind the amplified effects of oscillations of their model for motocycles in comparison to [45]. Especially since it happened only in the case of Tay Son street.

7. PLOS authors have the option to publish the peer review history of their article (what does this mean?). If published, this will include your full peer review and any attached files.

Reviewer #2: No

Reviewer #3: No

---

## [Author Response · Author response to Decision Letter 1]

20 Oct 2022

Reviewer #3: However, the authors need to pay careful attention to the following and revise the manuscript carefully before publishing can be done:

- Incomplete sentences e.g., line 333 page 10 "(Function :"

- Poorly formatted references in the text and poor latex formats e.g., line 300 page 10 "textbfchooseLane" and line 306 page 10 "see ??"

- Grammatical mistakes e.g line 117 page 4 e.g. having a bicycle or not

- Others

- Line 177 page 5, it is not clear if authors refer to the Table 1 in the article being referenced

- Need for consistency. Line 265 page 8. Compute_path written differently from other Java methods highlighted

These are just a few examples. Further proofreading is necessary for the whole manuscript.

Authors: Done.

Reviewer #3: Additionally, for the results it would be beneficial to readers if the authors include a brief explanation on the reason behind the amplified effects of oscillations of their model for motocycles in comparison to [45]. Especially since it happened only in the case of Tay Son street.

Authors: It is possible to observe more oscillations in our model than in that of [43] for motorcycles. One explanation is that the vehicles in the model of [43] are not based on lanes but on free spaces to move. Therefore, they have more possibility to execute maneuvers to avoid the vehicles stopped in front of them and therefore, if they decide not to stop at the red light (probability to stop at the red light of 0.5), to continue their way allowing less impact of the red light. In our model, the blocking of all lanes at the red light happens more quickly, leaving a more pronounced oscillation phenomenon. 

We do not find this phenomenon for Chua Boc because the road is much less wide than Tay Son, so even if vehicles have more ability to avoid obstacles, the blocking of spaces at the level of red lights happens much faster and with the phenomenon of oscillation.

---

## [Editor Report · Decision Letter 2]

30 Jan 2023

Dealing with mixed and non-normative traffic. An agent-based simulation with the GAMA platform

PONE-D-22-04270R2

Dear Dr. Taillandier,

We’re pleased to inform you that your manuscript has been judged scientifically suitable for publication and will be formally accepted for publication once it meets all outstanding technical requirements.

Kind regards,

Nan Zheng

Academic Editor

PLOS ONE
---

## [Editor Report · Acceptance letter]

24 Feb 2023

PONE-D-22-04270R2 

Dealing with mixed and non-normative traffic. An agent-based simulation with the GAMA platform 

Dear Dr. Taillandier:

I'm pleased to inform you that your manuscript has been deemed suitable for publication in PLOS ONE. Congratulations! Your manuscript is now with our production department. 

Kind regards, 

on behalf of

Dr. Nan Zheng 

Academic Editor

PLOS ONE